# Scientific literature on carbon dioxide removal revealed as much larger through AI-enhanced systematic mapping

Sarah Lück[1] ✉, Max Callaghan[1], Malgorzata Borchers[2], Annette Cowie[3], Sabine Fuss [1,4], Matthew Gidden [5,6,7], Jens Hartmann[8], Claudia Kammann[9], David P. Keller[10], Florian Kraxner [5], William F. Lamb [1,11], Niall Mac Dowell [12], Finn Müller-Hansen[1], Gregory F. Nemet[13], Benedict S. Probst [14,15,16], Phil Renforth [17], Tim Repke[1,18], Wilfried Rickels [19], Ingrid Schulte[1], Pete Smith [20], Stephen M. Smith [21], Daniela Thrän[2], Tiffany G. Troxler[22], Volker Sick [23], Mijndert van der Spek [17] & Jan C. Minx [1,11] ✉

Carbon dioxide removal plays an important role in any strategy to limit global warming to well below 2 °C. Keeping abreast with the scientific evidence using rigorous evidence synthesis methods is an important prerequisite for sustainably scaling these methods. Here, we use artificial intelligence to provide a comprehensive systematic map of carbon dioxide removal research. We find a total of 28,976 studies on carbon dioxide removal—3–4 times more than previously suggested. Growth in research is faster than for the field of climate change research as a whole, but very concentrated in specific areas—such as biochar, certain research methods like lab and field experiments, and particular regions like China. Patterns of carbon dioxide removal research contrast with trends in patenting and deployment, highlighting the differing development stages of these technologies. As carbon dioxide removal gains importance for the Paris climate goals, our systematic map can support rigorous evidence synthesis for the IPCC and other assessments.

To comply with the Paris agreement and to limit global warming well below 2 °C, rapid and deep GHG emissions reductions, need to be complemented with Carbon Dioxide Removal (CDR), potentially at the gigaton scale by the mid-century and beyond[1,2].

CDR has three distinct roles in climate change mitigation[1]: first, to reduce net $CO_2$ and greenhouse gas emissions in the near term, specifically in the land sector; second, to offset residual emissions from "hard to mitigate" sectors like industry, long-distance transport, and agriculture in the medium term[3,4]; and third, to support sustained net-negative emissions in the long term, helping to lower global temperatures in overshoot scenarios and stabilising warming at or below 1.5 °C[1]. Of course, CDR cannot compensate for stringent emission reductions, which need to be prioritised even in hard to mitigate sectors[5]. There are also deep uncertainties with respect to how fast

CDR can be sustainably scaled-up, and whether the reversal of temperature overshoot can be safely achieved[6]. This underlines the need to reduce emissions as fast as possible, while providing sufficient policy support that CDR can actually deliver gigatons of removals in the second half of the 21st century[2,7].

CDR has been a key part of climate change mitigation discussions in the scientific literature, but has often been separated into distinct knowledge domains. A stream of literature going back to the first IPCC assessment reports has considered the potential contributions of enhanced natural sinks through afforestation or soil carbon sequestration to achieve net emissions reductions[8,9]. This area has since broadened to include analogous nature-based approaches in other ecosystems, such as coastal blue carbon[10], alongside options aimed at enhancing ecosystems' ability to absorb and store $CO_2$, like ocean

fertilisation and macroalgae afforestation[11,12]. Bioenergy with carbon capture and storage (BECCS) technologies have gained prominence in the early 2010s as an explicit option for achieving negative emissions in the integrated assessment modelling (IAM) literature[13–15]; while a range of other technologies such as biochar produced by pyrolysis, direct air carbon capture and storage (DACCS), enhanced weathering (EW), and ocean-based approaches such as ocean alkalinity enhancement (OAE) are now gaining more scientific attention[15–17].

In the policy domain, CDR has gained increasing attention in recent years[18,19], but many countries still lack concrete policies to scale CDR[2]. This has led to a considerable gap between countries' (so far limited) plans to develop and deploy CDR versus CDR's estimated role in mitigation scenarios that stabilise global temperatures at an increase well below 2 °C[2,19–21]. One of the challenges here is that there is a large spread of possible CDR levels that countries might aim for, in part driven by model assumptions of technological innovation and potential market adoption[22].

In the age of big literature[23,24]—where the scientific literature grows at increasing rates—balancing a research question's scope and the resource demands for reviews, like reviewer time, is increasingly challenging[25]. To address this issue, systematic mapping methodologies (systematic maps, evidence gap maps, etc.) have been developed by the evidence synthesis community[26,27], to map existing literature, identify knowledge gaps and clusters, and guide where reviews are most beneficial. However, these methods remain resource-intensive, prompting discussions about the prospects of automation[28,29]. Proposals for implementing such automated synthesis approaches have been developed across various scopes and scales[23,30–32].

There is currently little systematic oversight of the available CDR literature. As the IPCC's 7th Assessment Cycle is starting and CDR-related policies and targets are being established, it is timely to assess the current landscape of evidence for CDR. Previous research suggests that there is a large and fast-growing evidence base on CDR, but the few available overviews of the field have rapidly become outdated[33,34], only give a coarse overview[2,35] or are limited in scope by manual efforts supported by community-crowdsourcing[36].

The diverse range of CDR options and multidisciplinary fields involved in CDR research also adds to the complexity of this task, as researchers from different disciplines, each with their own specialised languages and methodologies, may be working on the same issues without fully knowing or engaging with each other due to misaligned terminology. It is also crucial to identify and keep track of gaps in the literature in order to effectively allocate research resources.

Here, we follow a systematic mapping methodology to comprehensively lay out the body of knowledge on CDR. We ask an open-framed question—"what is the available evidence on CDR?"—and follow a robust, stepwise methodological procedure that ensures transparency, comprehensiveness and reproducibility. Traditionally, systematic maps have been compiled manually and therefore are often limited in scope. Here, we use an approach that deploys machine learning methods to automate labour intensive tasks to provide an assessment at scale. By doing so, we are not only able to quantify the size and scope of the research landscape of CDR and its temporal dynamics, but are also able to assess the distribution of research efforts across various dimensions, including CDR options, research methodology, disciplinary structure and geographic focus. Furthermore, our machine-learning approach enables swift updating of the dataset in the future. Given the growing importance of CDR in the context of net-zero strategies and temperature overshoot, our publicly available database of CDR research will be of benefit to the research community as well as upcoming scientific assessments of CDR.

In this article, we first quantify the total volume of CDR literature and examine its temporal trends, as well as dissecting the literature by individual CDR options to highlight shifts in research focus. Next, we investigate the origins of these studies, exploring regional profiles and analysing research that specifies geographic locations to identify patterns in CDR research distribution. We then assess the focus of the studies, including the scientific methods employed, to understand how research approaches have evolved. Additionally, we evaluate the representation of CDR literature in the recent IPCC report, comparing it to the overall CDR literature to highlight any discrepancies. Finally, we compare the attention given to different CDR options across various contexts, including Integrated Assessment Model (IAM) scenarios, deployment strategies, and investment patterns.

## Results
### Literature on CDR is much larger than previously estimated
There is a much larger body of CDR research than previously suggested. Based on our machine learning assisted approach that enables us to identify CDR studies with high precision ($0.88 \pm 0.0119$, meaning the proportion of relevant studies among those identified is high) and recall ($0.93 \pm 0.005$, indicating most relevant studies are captured)—see "Methods", Supplementary Methods 3 and 4 and Figs. 1 and 2—we predict a total of $28,976 \pm 3800$ scientific studies in the Web of Science and Scopus (the two largest bibliographic core collections). This is 3–4 times larger than what previous scientometric studies[33] or ongoing community efforts to manually track CDR research[36] have suggested when comparing the same time range. For the former study, this discrepancy likely arises from their reliance on non-machine learning methods, which forced a high-precision, low-recall search approach. In the case of the manual tracking efforts, the rapid expansion of CDR literature has simply made comprehensive tracking unfeasible.

CDR research today comprises only 5% of the overall scientific literature on climate change[23], but growth in CDR research is faster than for climate overall. We observe an average annual growth rate of 17% over the past ten years compared to a 12% growth rate for the literature on climate change (Fig. 3).

### Patterns of CDR research are uneven and dominated by biochar studies
The distribution of research across different CDR options is highly concentrated on biochar and land-based methods such as soil carbon sequestration and afforestation/reforestation. Most growth rates of the different CDR options are generally higher compared to the climate change literature as a whole (Fig. 3 and Supplementary Table 1).

Biochar research is covered in 56% of the 28,976 scientific publications on CDR. Considering 2022 only, the share of publications covering biochar even increases to 62%.

With an annual growth rate of 18% in the past 5 years and its large number of publications, biochar is the main driver of the high growth rate of the entire CDR literature. The second largest category, SCS, with 24% of the total literature, is also growing fast at 14% per year in the past 5 years.

Other biological CDR options make up a sizable amount of the CDR literature, such as afforestation/reforestation with 12% of all studies, agroforestry with 9.7% of all studies, coastal wetland (blue carbon) management with 4.7% and landscape restoration, such as peatlands with 3.5%. Growth rates in the past 5 years are generally in the range of 20–22%, except for the long established CDR options such as afforestation/reforestation and forest management with 14% and 15% respectively. Growth in newly emerging research areas tends to be particularly high, as initial literature numbers are low, making each new publication a relatively larger addition to the existing body of work.

BECCS is represented in only 5.6% of all studies on CDR, despite being the most common CDR option in most scenario pathways for meeting the Paris temperature goal[37–39]. DACCS accounts for only 2.8% of all CDR studies. The annual growth rate of BECCS is volatile and the average of the past 5 years is 12%.

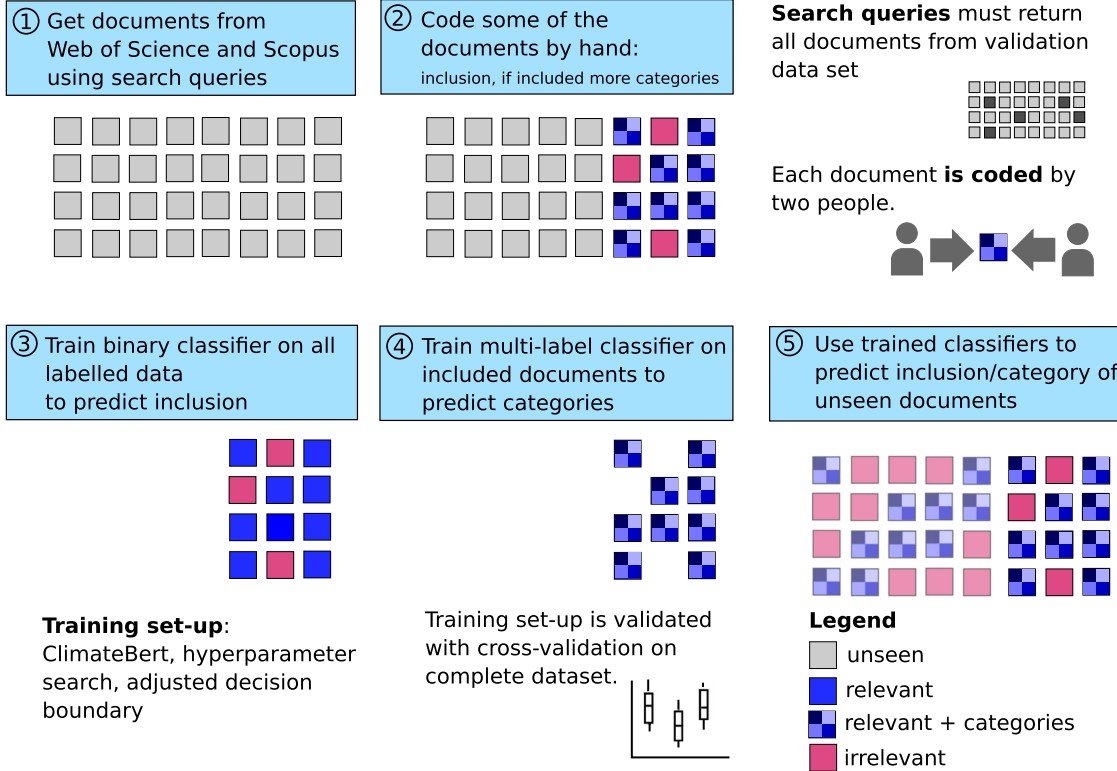

**Fig. 1 | Overview of the data retrieval for this study.** Squares symbolise documents, a coloured square a document with labels, either assigned by hand (solid colour) or automatically (faded colour). Red documents are excluded, blue ones included. Step 1: 70,000 documents were retrieved from databases using search queries. Step 2: Of these about 6000 documents are sorted (=coded) by hand into being on CDR (relevant, blue squares) or being not on CDR (irrelevant, red squares). Documents on CDR were additionally described with CDR options, see Fig. 2, and other categories. Steps 3 and 4: The relevance labels and additional categories were used to train machine learning classifiers. Step 5: The trained classifiers were used to extend all labels to the unseen ~64,000 documents. Detailed information on methods can be found in the "Method" Section and the Supplementary Methods 3 and 4.

Other CDR options are much less represented in the scientific literature: for ocean fertilisation, EW and OAE, we found less than 50 studies for each option per year.

## CDR research is concentrated in China and OECD countries

We use the first author affiliation to infer the origin of the studies. This approach simplifies the complexities of international collaborations, where authorship, lead roles, and funding often span multiple countries. However, we consider it a valuable proxy for drawing meaningful conclusions about the research origins. With our approach, we find that China is responsible for the largest amount of research on CDR with 6452 studies (30% of all studies where author affiliation is available), followed by the United States (2667 studies, 13%) and the United Kingdom (953 studies, 4%). Only 3.4% of all studies with author affiliation have a first author affiliation from South America and 2.8% come from Africa, see Supplementary Fig. 6.

Though global research trends favour land-based CDR options, specialisation varies. China focuses more on biochar research, Europe on BECCS, and North America, particularly the US, on DACCS in comparison to the average shares per CDR option across all countries. Research on ocean-based CDR options is more predominant in Oceania and North America (Fig. 3). We provide detailed country profiles for CDR research in the Supplementary Fig. 5.

Roughly one third of CDR research refers to specific geographic locations, identified through named entity recognition in titles and abstracts[40]. Place-based research is important for evaluating CDR implementation in situ, including aspects such as effectiveness of removing $CO_2$ and environmental or social side-effects. Out of 28,976 studies, 9305 mention a location, of which 74% are countries and 25% are sub-national locations such as federated states, counties

or cities. Soil and vegetation-based CDR options feature more place-specific research, with afforestation/reforestation studies at 65%, compared to 33% for enhanced rock weathering, and only 10% for DAC(CS) research, Fig. 4. Further information on regional based research with details to the specific regions can be found in the Supplementary Figs. 7 and 8.

## CDR literature focuses on technology research using experimental methods and modelling

Our ML approach further enabled us to classify CDR research contents along key dimensions. In particular, we used our classifiers to distinguish research methods and the broad area of research. Additionally, we used the journal, in which a publication appeared, to determine academic disciplines in line with the relevant OECD Category scheme[41].

We refer to CDR research that aims to understand, design or further develop CDR options, their efficiency and side-effects as "technology research". As indicated by Supplementary Fig. 10, technology research accounts for about 89% of all studies across all individual CDR options. We refer to survey or focus group research on public perceptions and attitudes to CDR as "public perception" (0.8% of the total), and integrated assessment scenario research as "socioeconomic pathways" (9% of the total). We further classify "policy and governance" research, and studies on the "earth system" that evaluate global carbon cycle or land aspects of CDR implementation, see for example[42,43], even though these categories remain relatively rare (at 3.8% and 0.6%, respectively).

In general, patterns of research vary across the technology categories. The literature on CDR in general features mainly policy and governance (28% of papers on CDR in the general category) as well as integrated scenario research (45%). We also find larger shares of

## Overview of classes and classification process for CDR options

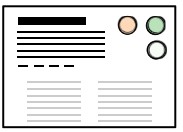

All classifications only based on title, abstract, author keywords

A document can be assigned to multiple categories.

General literature on CDR
Studies without a limited focus on one CDR option or mentioining multiple CDR options.

**BECCS**
= **B**ion**e**rgy production with **C**arbon Dioxide **C**apture and **S**torage technology.

**DACCS**
= **D**irect **A**ir **C**arbon **C**apture and **S**torage. Chemical process by which CO2 is captured directly from the ambient air, with subsequent storage.

**Soil Carbon Sequestration**
Land management changes which increase the soil organic carbon content.

**Afforestation / Reforestation**
Planting new forests on non-forested land and replanting cleared forests.

**Restoration of Landscapes/Peat**
Formerly destroyed landscapes that sequester CO2 are restored to their natural conditions.

**Agroforestry**
Land use management system in which trees or shrubs are grown around or among crops.

**Forest Management**
Every aspect of management which leads to a sustained carbon dioxide removal.

**Enhanced Weathering (land-based)**
Dissolution of silicate and carbonate rocks by grinding these minerals to small particles and applying them to soils.

**Ocean Alkalinity Enhancement**
Dissolution of silicate and carbonate rocks by grinding these minerals to small particles and applying them to oceans.

**Blue Carbon Management**
Blue carbon is carbon captured by living organisms in coastal and marine ecosystems, and stored in biomass and sediments.

**Ocean Fertilisation & Artificial Upwelling**
OF: Nutrient supply to the near-surface ocean to enhance biological production
UW: Pumping of nutrient-rich deep ocean water to the surface where it has a fertilizing effect.

**Biochar**
Production of biochar, a stable carbon-rich material formed by heating biomass in a low-oxygen environment, with use as a soil amendment or in the built environment.

**Fig. 2 | Schematic overview of the coding guidelines used for inclusion in the systematic map.** The definitions of each carbon dioxide removal (CDR) option shown in this figure served as the baseline for inclusion. Additional coding guidelines are provided in the Supplementary Methods.

scenario research for some individual CDR options—particularly for BECCS (31%) and forest-related CDR options (21–22% for Forest Management and Afforestation/Reforestation), which were the first to be implemented in the modelling community[38,44].

CDR research is published to a large extent in journals with a natural science or engineering focus and tends to be rooted in experimental and modelling study designs. In particular, 50% of the studies are published in natural sciences, 26% in agricultural sciences and 22% in engineering and technology journals (see Fig. 5). Only 3% of the publications are published in journals with a focus on social science, including economics.

Research designs vary substantially across CDR options, but experiments, reviews and modelling studies are most common. Overall, 86% use experimental methods, either laboratory (48%) or field (38%) experiments, driven mainly by research on biochar and soil carbon sequestration. Reviews (21%) and modelling (18%) make up another large proportion. However, certain research designs are more dominant in the literature for specific CDR options. For example, field studies typically make up a substantial share of forest-based CDR options, but also blue carbon and ocean fertilisation, while laboratory experiments, i.e., experiments in a controlled environment[45], are dominant for biochar, soil carbon sequestration, but also some engineered CDR options such as DACCS. Interestingly, BECCS studies to date focus strongly on modelling, highlighting their prominent role in climate protection scenario work. Across all CDR options, reviews are widely available—from 11% for forest management up to 32% for OAE, and 34% for the literature on CDR in general (Fig. 4).

**IPCC reports differ greatly from scientific literature**
Next, we analyse how the research landscape is reflected in the most recent 6th Assessment Report by the IPCC. For this, we extract all

citations from the IPCC AR6, all working groups, and identify those studies which are present in the literature on CDR by matching titles. Although it is clear that the IPCC cannot assess all of the large and growing body of available research[33], it is essential to understand which main topics are emphasised or overlooked. We also acknowledge that differences between the two literature bodies can arise from various factors and that the main topic distributions should not necessarily align—as Hume remarked, what "is" doesn't necessarily lead us to what "ought to be".

We find that IPCC assessments are not a broad reflection of attention patterns in the underlying scientific literature on CDR options (Fig. 6). Overall, only a small fraction (2% of the CDR literature) of CDR studies are directly assessed. While the IPCC includes a relatively higher proportion of reviews (19% vs. 15%) and systematic reviews (3% vs. 1%) compared to the overall CDR literature, we believe incorporating even more of these could further enhance its ability to fulfil a stated goal of the IPCC—which is to comprehensively evaluate the available evidence.

IPCC assessments cite a wide range of CDR options, but are predominantly concerned with BECCS (27%)—probably due to its prominence in climate change mitigation scenarios (see Fig. 6)[2,46]. The major focus on biochar in the research community is not reflected in IPCC citation patterns.

The fact that IPCC assessments have tended to focus on scenarios is underlined by an observed shift from experimental research (86% of CDR research; 10% of IPCC citations) to modelling work (13% of all CDR research; 37% in IPCC citations) and data analysis (10% of all CDR research, 19% in IPCC citations). The focus on technology research in the literature (89% of all CDR research; 44% in IPCC citations) is replaced by much more prominence of scenario work (socio-economic pathways) (9% of all CDR research; 33% in IPCC citations) as well as

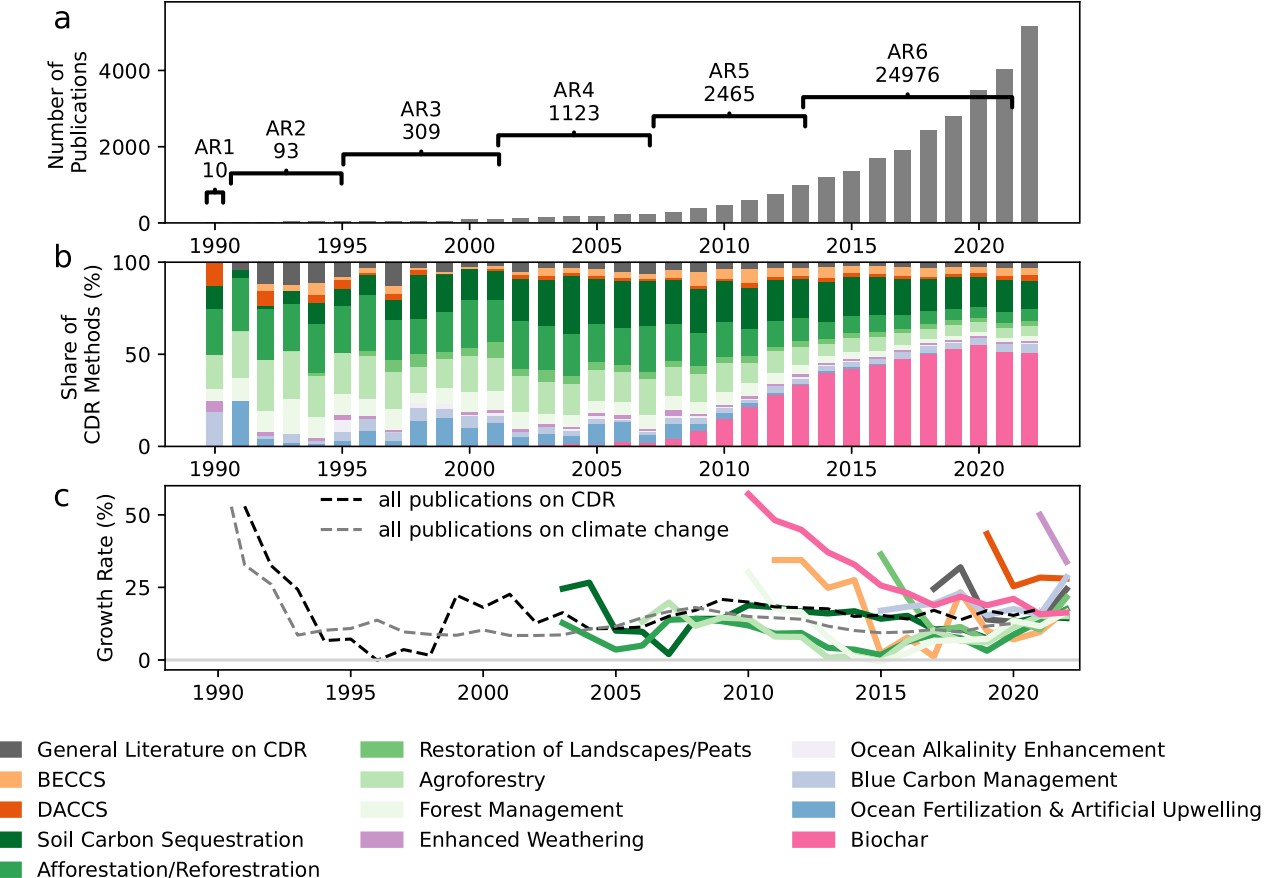

**Fig. 3 | Time development of the scientific literature on CDR in Web of Science and Scopus. a** Total number of publications per year between 1990 and 2022. Additionally, we note the number of publications released during each Assessment Report (AR) cycle of the Intergovernmental Panel on Climate Change, the latest AR6 considered publications until 2021. **b** Share of CDR option covered in scientific publications. Multiple options per publication are possible. A more complete list of all counts per option is published in the Supplementary Fig. 1. **c** Annual growth rate of the scientific literature on CDR, climate change and individual options. Growth rate is only calculated if there were more than 50 publications in total available. Colorblind-friendly versions of the middle and lower panel can be found in the Supplementary Figs. 2 and 3. Source data is provided as a Source Data file.

research on policy and governance (4% of all CDR research; 22% in IPCC citations). All this reflects that IPCC assessments focus on the exploration of alternative scenarios with different climate outcomes, societal development pathways and mixes of mitigation strategies, intended to inform policy development[33,47,48].

### Shares of CDR options vary across indicators of policy and practice

Finally, we find that the CDR options being researched most intensively are not the ones being most actively deployed, developed or invested in (Fig. 7, ref. 2, Chapter 3,6,7). Again, we do not imply that these distributions should necessarily be similar; rather, we aim to highlight and reflect on the differences between these categories. For example, while CDR research strongly focuses on biochar and soil carbon sequestration, the vast majority of current deployment (2Gt yr-1 or 99.9%[49]) is from afforestation and reforestation. Conversely, even though only 2Mt yr-1 of $CO_2$ removal is currently delivered by more novel CDR options—mainly BECCS (78%) and biochar (21%)[2] —these technologies receive an enormous amount of scientific attention or are widely discussed in the scenario literature. Similarly, about 80% of the CDR patents are for BECCS and DACCS[2,50]. 75% of announced investments in CDR focus on DACCS projects[51]. In long-term mitigation scenarios that achieve the Paris long-term temperature goals[52] mainly BECCS (99%), afforestation (67%) and DACCS (29%) are the CDR options included. There is not a single scenario dealing with biochar or

soil carbon sequestration due to a lack of implementation of these CDR options despite their potential co-benefits, such as food security or $N_2O$ emission reduction[53].

## Discussion

In this article, we provide a comprehensive evidence map of the CDR literature. Our machine learning assisted approach follows a systematic mapping methodology[26,27], and automates key labour-intensive parts of the process[28]. This allows our systematic map to cover the entire research domain around CDR rather than being limited to a niche area of literature due to resource limitations[29]. As a result, we were able to quantify the CDR research landscape in an unprecedented way. Moreover, the automated classification can also be applied to newly published CDR research, representing a critical step forward in accelerating learning on CDR and providing high-quality evidence syntheses on the topic. This is particularly important as we continue to face a rapidly growing evidence base.

At the heart of our map of CDR research is a classification system trained with about 5300 manually labelled documents that is able to predict not only if a scientific publication is relevant for the evidence map, but also the CDR option, the broad area of research it is situated in, and the research methodology applied. This literature base serves as a foundation for further analysis and can be easily expanded with additional features that provide more detailed descriptions of the scientific literature. In this context, a simple keyword search can enrich

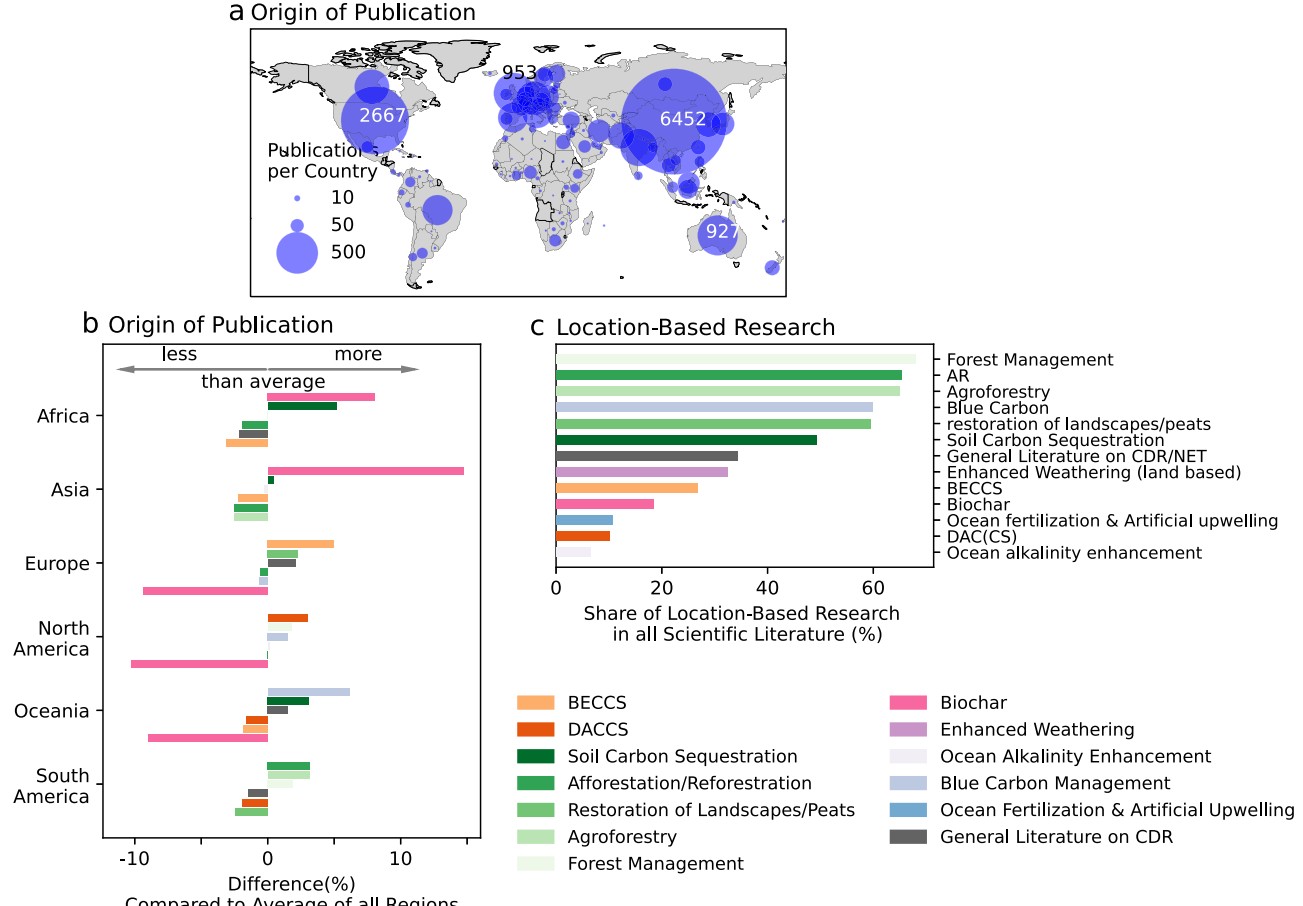

**Fig. 4 | China dominates the scientific literature on CDR. a** Number of studies per country based on first author affiliation. The three highest study counts are added. **b** We sort the origin of the study into the world regions. For each world region, we compare the percentage difference of the investigated CDR options against all others from the complete dataset. Displayed are only the three highest and the three lowest differences. **c** Location-based research derived from locations mentioned in title and abstract. Displayed is the share of location-based research in all scientific literature per CDR option. A colorblind-friendly version of panel b can be found in the Supplementary Fig. 4. Source data is provided as a Source Data file.

the literature landscape across a diverse range of interests. All documents together with their categories can be downloaded from our literature hub: climateliterature.org/#/project/cdrmap.

We find that the CDR literature is 3–4 times larger than previously estimated[33,36] when comparing the same time frame. The reason for this is that our machine learning assisted approach enables us to be systematic in our procedure and at the same time achieve both high levels of precision and recall. Previous approaches had either designed precise search strings that lack recall[33] or relied on manual tracking of the field, which has simply grown too large[36].

While our CDR map represents the most comprehensive work in this area to date, it does not offer a complete portrayal of CDR science. Our search, focused on English-language articles in Web of Science and Scopus, overlooks significant portions of literature in other languages and grey literature, particularly relevant for emerging technologies like BECCS and DACCS. Estimates suggest that Web of Science captures only about 40% of scientific publications[54], implying potentially another 50,000 CDR-related publications. Additionally, the large opportunities for CDR functionality that are provided by converting captured $CO_2$ into long-lived economically viable products (Carbon Utilization Infrastructure, Markets, and Research and Development 2024) has not yet been implemented in this review but is subject of ongoing work.

Our machine learning classification system is not perfect and varies in accuracy across tasks. For example, while we are able to

predict biochar with a F1-score of 0.98, our classifiers perform much poorer for classifying agroforestry. However, our supervised machine learning procedures involve in-depth validation and as such, we establish transparency about our uncertainty in quantifying the evidence base—something rarely provided in manually compiled evidence maps, which are commonly viewed as gold standard.

Here we confirm previous research[33] that the expansion of the scientific literature on CDR is taking place more rapidly than for climate change as a whole. Overall, we find that CDR research is highly concentrated on particular CDR options, specific areas of research as well as research approaches. The CDR literature is dominated by biochar research today—with a geographical centre in China. This development is relatively recent and driven by much higher publication rates than observed for any other CDR option. There could be a number of drivers that explain the large uptake of biochar research in China, including institutional developments (e.g., increased core funding at agricultural universities, publishing incentives, or research grants), strengthening scientific networks (e.g., new societies, journals, project collaborations and exchanges), or a concerted push from the policy sphere (e.g., strategic research funding, support for public-private enterprises). Of course, applied research cannot be abstracted from its surrounding geographic and economic contexts. It is therefore not unexpected to find CDR research niches in different contexts (e.g., biochar in agriculturally productive regions, ocean-based CDR in coastal regions, DACCS and BECCS in industrialised regions).

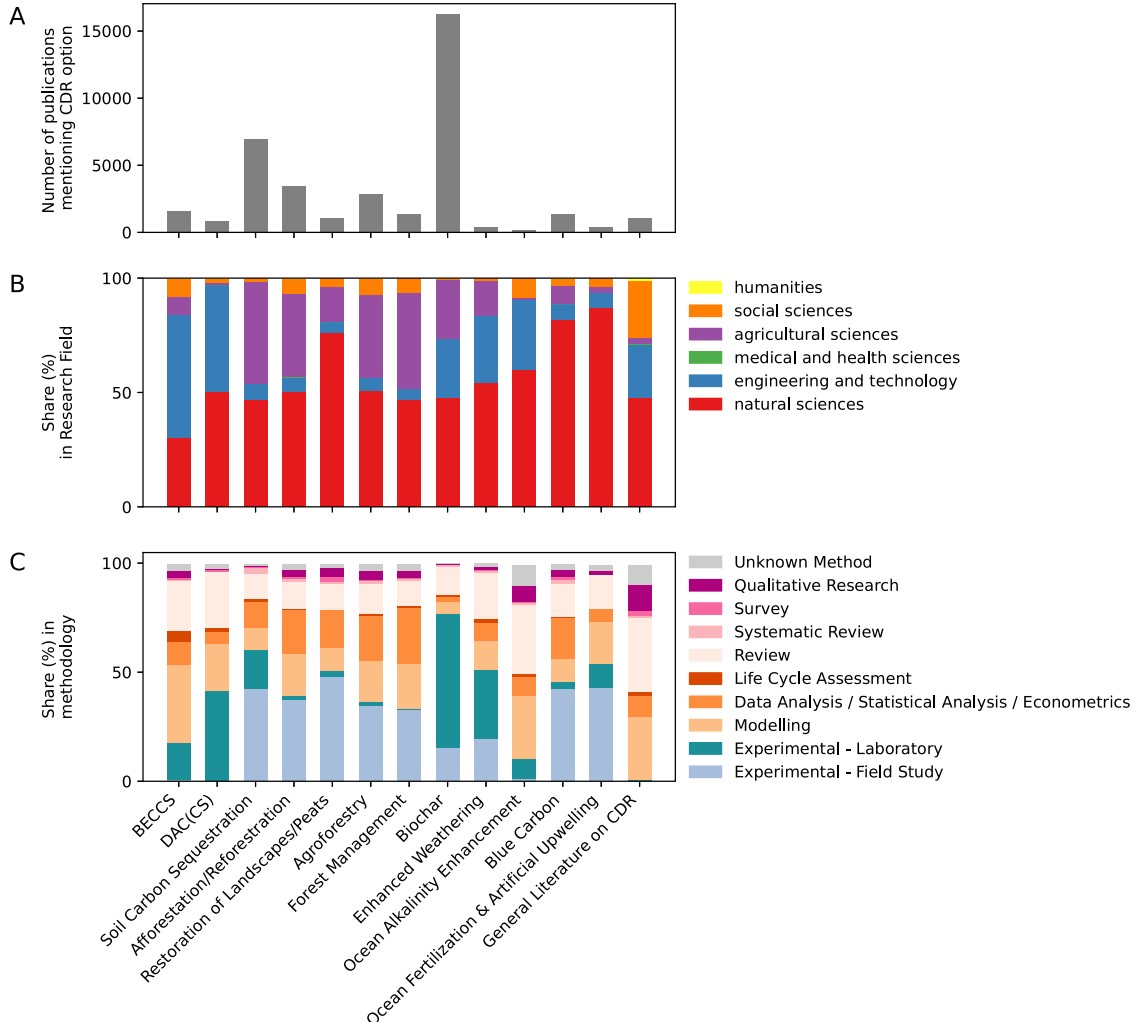

**Fig. 5 | Most studies focus on investigating the CDR option from a technical perspective, where the technology or ecosystem management method itself is investigated. a** The number of studies which report on each of the CDR options. One study can report on multiple CDR options. **b** For each CDR option the share of research fields the studies were published in. This is based on meta-data from the Web of Science and follows the OECD Category scheme[41]. **c** For each CDR option the share of scientific method used in the studies as identified by our classifier. One study can use multiple methods, see Supplementary Table 3. Source data is provided as a Source Data file.

Patterns of research are also distinctly different from what we observe in policy and practice. In part, this reflects the differing technological readiness levels of each CDR option, which vary from early stage research (e.g., enhanced weathering), to pilots and demonstrations (e.g., DACCS), and full-scale commercialisation (e.g., afforestation/reforestation)[55]. This may explain why—compared to the available research—patenting and investment activity has been relatively more active for DACCS, where a series of recent demonstrations have taken place. The tendency for scenarios to include a very significant share of BECCS also reflects path dependencies in model development, which already started to implement this technology option in the 2010s. It should be noted, though, that there are active developments to expand the range of CDR options in IAMs[56]. CDR deployment is also driven by issues like social acceptance, where methods with higher perceived "naturalness" and a longer history of practice (e.g., afforestation/reforestation) have a clear advantage[57].

We show that IPCC assessments do not reflect publication patterns in the underlying scientific literature. Systematic mapping efforts can help identify topical areas worthy of focus, and these may need to be adopted into assessment procedures. Indeed, a first practical step is to identify, evaluate and utilise the existing body of reviews and

metastudies, which too have been under-cited in the IPCC in favour of a limited set of primary studies.

We identify a few "evidence hubs" where systematic reviews, i.e., a complete and robust assessment of the available literature, would be feasible based on the current body of work. For example, the extensive literature surrounding afforestation policies offers an opportunity for an ex-post analysis to yield insights on the long-term effectiveness and social impacts of these efforts. Updated assessments of CDR costs and potentials are also needed—expanding beyond previous efforts, such as those by ref. 37—to reflect recent advancements in the underlying evidence base. Additionally, there is sufficient evidence to support a systematic review on monitoring, reporting, and verification, as identified by ref. 58, which is instrumental to developing reliable certification schemes for CDR.

Finally, in terms of evidence gaps, we note less of a research focus on ocean alkalinization, EW, and agroforestry. We also observe there are few studies on more novel CDR options, such as DACCS, that are place-specific. More localised research is needed, given that the successful implementation of these methods is often dependent on local geographies (e.g., geological reservoir access) and socio-economic contexts (e.g., social acceptance, energy prices and availability). Additionally, our results point to a need for more research on CDR in

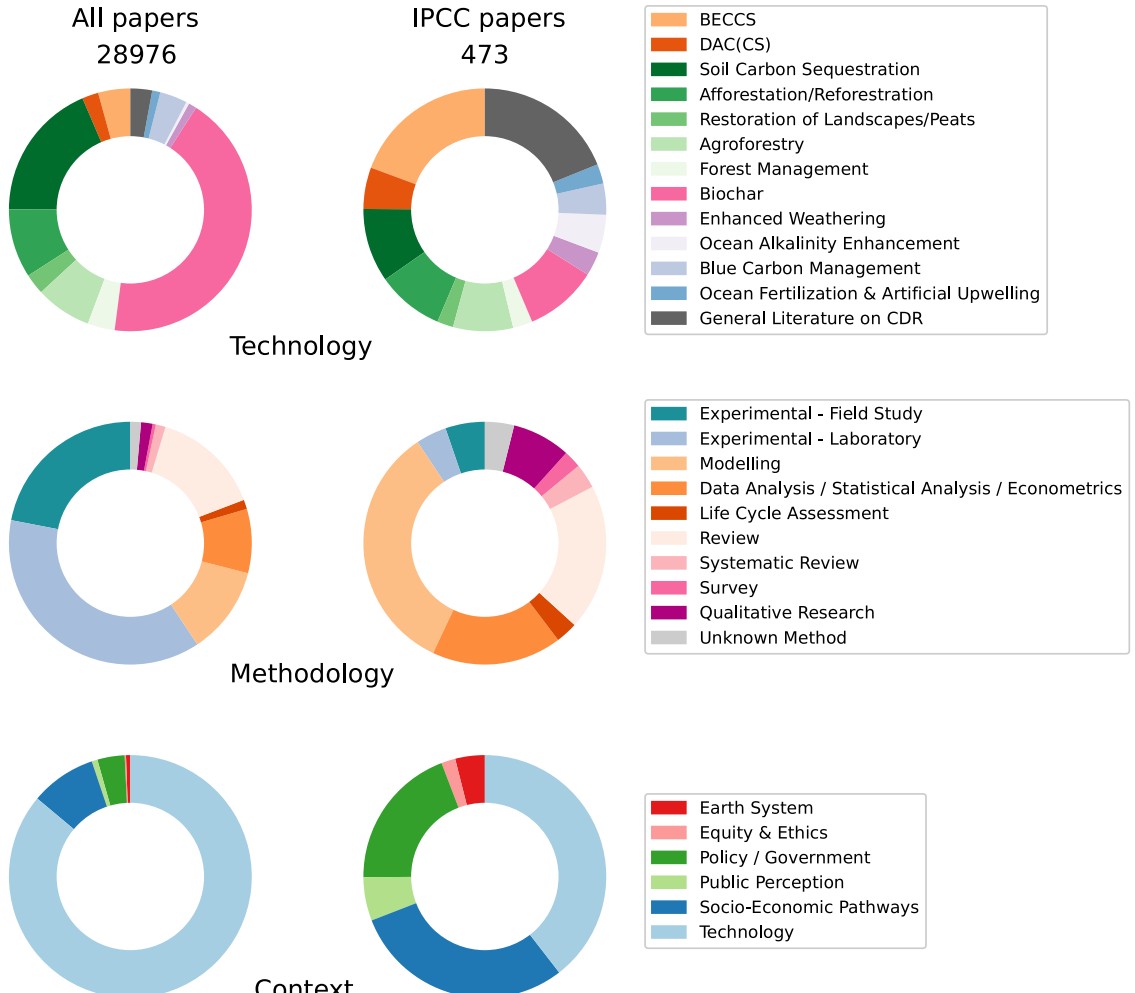

**Fig. 6 | Comparison of all literature on CDR with CDR papers cited by the 6th assessment of the IPCC (reports of all three working groups).** Reading guide for colorblind people: The categories follow the order listed in the legend, beginning from the top of each circle and proceeding counterclockwise. Source data is provided as a Source Data file.

the social sciences and humanities, for example, to support evidence-based decision-making on questions of governance and equity. This type of research will be increasingly important as focus shifts towards the implementation of CDR at scale and policy design to support this, as is implied by the ambition of net-zero targets in the context of relatively slow action of mitigation policies.

## Methods

### Systematic map - protocol
We use an approach assisted by machine learning to provide the a comprehensive evidence map of CDR research. We follow the well established guidelines for systematic mapping[25], wherever possible, and adjust them as needed to align with our machine learning approach. We document all steps in a detailed systematic map protocol for transparency and reproducibility[45], which is summarised in Fig. 1 and Supplementary Fig. 11.

### Document search
We started by developing, for each CDR option, search strings with high levels of recall to make sure that as few scientific articles are missed as possible. The search strings include keywords describing the CDR technology, see Supplementary Information for the full search queries. For long established CDR options, such as afforestation, we included keywords that make sure the CDR option is evaluated with a focus on carbon sequestration. The development of search strings was done iteratively by validating against an independent list of publications on the various CDR options ensuring that all documents are returned. The validation dataset was extracted from IPCC AR6[59,60] and 50 randomly selected publications from the CDR bibliography[36] published by the Climate Protection and Restoration Initiative. The search queries are available in the Supplementary Table 3. We then ran the final search strings on Web of Science and Scopus on March 28th, 2022 and May 3rd, 2023 and retrieved 75,518 bibliographic records after de-deduplication. Further information on this procedure and information on the validation dataset is available in Supplementary Table 1 and Supplementary Method 1.

### Document relevance through machine learning
In the next step, we work towards precision by developing a machine-learning classifier to distinguish relevant, namely all studies on negative emissions and CDR, from irrelevant scientific studies in our query. We manually screen and annotate a total of 5339 documents— 100–600 per CDR option—if they should be included in the map (distinction between blue and red squares in Fig. 1) according to our codebook. To ensure reproducibility[61,62], each document is screened and annotated by two coders as recommended by the relevant guidelines[25]. We use our annotations to train and validate binary classifiers, i.e., automatic sorting

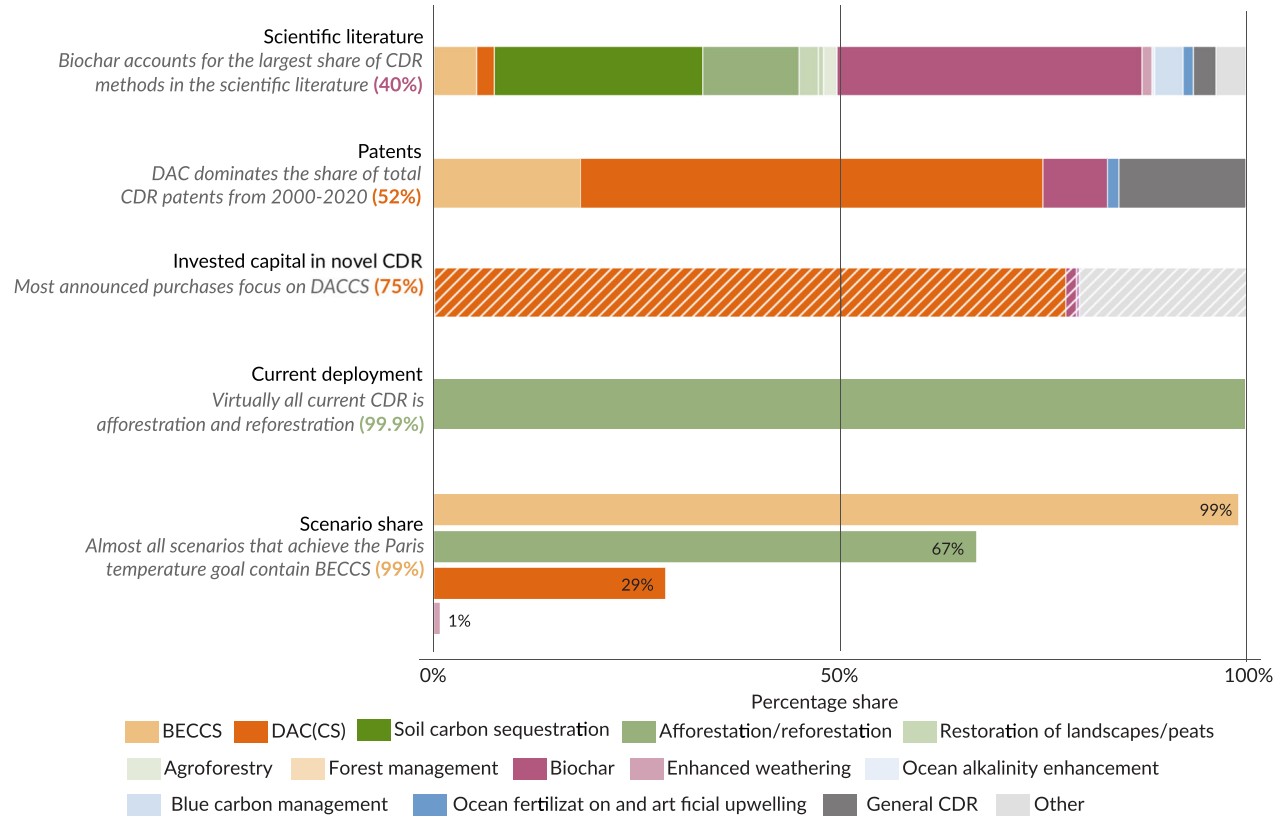

**Fig. 7 | Share of CDR options for current deployment of CDR, patenting activity, scientific literature, invested capital and considerations in the scenarios assessed in the recent IPCC report.** Data was taken from ref. 2. Source data is provided as a Source Data file.

into predefined categories, to predict inclusion, using the title and abstract of the documents as inputs. The best performing classifier (F1: 0.91; ROC-AUC: 0.85) is derived from ClimateBERT— a transformer-based pre-trained language model, which has been fine-tuned to better represent domain-specific language used in the climate change context, including in scientific abstracts[63]. Further details and an explanation of our model validation procedure are available in the Supplementary Methods 2 and 3.

**Document classification through machine learning**
We further annotated all relevant scientific articles from our manually coded training and validation set with regard to the CDR options covered (Afforestation/Reforestation, Restoration of landscapes/peats, Agroforestry, Soil Carbon Sequestration (SCS), Blue Carbon Management (mangroves, macroalgae, seagrasses, and salt marshes), EW, OAE, Ocean Fertilisation/Artificial Upwelling, Bioenergy Carbon Capture and Sequestration (BECCS), Direct Air Carbon Capture and Sequestration (DACCS), Biochar, additionally we include General Literature on CDR with no focus on a specific technology), the scientific method used, as well as the broad area of research (technology study, policy & governance, equity, public perception, socio-economic scenarios, earth system science). Definitions of all CDR methods used to code the documents are shown in Fig. 2. Additional information on how we distinguished the different classes can be found in our coding protocol[45]. The additional categories are represented in Fig. 1 by the different blue shades for each annotated relevant document. We used these annotations to train three multi-label classifiers for second stage predictions, and apply them to documents predicted relevant at the first stage. We achieve Macro F1/Macro ROC AUC scores 0.77/0.87 for the "technology" classifier, 0.69/0.89

for the "methodology" classifier and 0.62/0.77 for the main "area of research" classifier.

**Machine learning validation**
Throughout this process, we evaluate and validate our methodological choices. We test our ClimateBERT classifications against classifications from DistilBERT[64] as well as a much simpler classification approach, where we use tf idf-encoding together with an SDGClassifier with Huber-loss[65]. ClimateBERT is chosen here due to its better performance (see Supplementary Table 3). We optimise classifier performance by tuning the hyperparameters of our model using the Python package RayTune[66]. Finally, we test the complete training strategy of all classifiers in a threefold cross validation providing us with comprehensive estimates of how the classifiers perform on the complete dataset (cf. Supplementary Table 4–6). To estimate the confidence intervals for absolute counts, we estimated the True Positive Rate and False Positive Rate from our validation procedure and calculated their confidence intervals using binomial proportion confidence intervals, see Supplementary Method 4.

**Locations in title and abstract**
To find the locations in title and abstract, we deployed the Python package Mordecai[40].

## Data availability
All documents, including their classification, are available for download on our literature hub at climateliterature.org/#/project/cdrmap. The interactive website allows users to search for documents and filter by category. Source data to Figs. 3–7 are provided as a Source Data file. The data generated in this study have been deposited in ref. 67. Source data are provided with this paper.

## Code availability

All code used for training the machine learning models and analysing the data is accessible at https://github.com/mcc-apsis/cdr-map[68].

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

## Acknowledgements

We thank Christiane Hamann, Doménica Michelle Jaramillo Sánchez, Ronja Kelch, Fariha Mawla, Fabian Metz, Leon Stephan and David Verdugo-Raab for their effort in coding most of the documents. S.L., M.C., F.M.H., W.L., T.R., M.G. were supported by the ERC-2020-SyG "GENIE" (grant ID 951542). I.S., S.F., T.R. and J.M. received further funding from the German Ministry of Research and Education under the CDR-SynTra project (01LS2101F), C.K. and J.H. under the CDRterra PyMiCCS project (Grant Refs: 01LS2109C and 01LS2109A). S.M.S. was supported by the CO2RE Hub, funded by the Natural Environment Research Council (Grant Ref: NE/V013106/1). D.P.K. was supported by the European Union's Horizon 2020 Research and Innovation Program under grant 869357 (the OceanNETs project) and the German Ministry of Research and Education CDRmare projects RETAKE, sea4soCiety, and ASMASYS. P.R. is supported by the UK's Industrial Decarbonization Research and Innovation Centre (EP/V027050/1). F.K. and W.R. received support from the EC Horizon Europe project UPTAKE (Project # 101081521). D.T. and M.B. received funding from the German Ministry of Research and Education under the BioNET project (01LS2107A). M.J.G. is also affiliated with Pacific Northwest National Laboratory, which did not provide specific support for this paper.

## Author contributions

S.L., S.F. and J.M. conceptualised the work. S.L., M.C. and J.C.M. contributed to methodology. SL did the investigation and visualisation. S.F., J.C.M., M.C. and T.R. supervised the work. Writing–original draft: J.M. and S.L. wrote the original draft. M.C., M.B., A.C., S.F., M.G., J.H., C.K., D.P.K., F.K., W.L., N.D., F.H., G.M., B.P., P.R., T.R., W.R., P.S., I.S., S.M.S., D.T., T.G.T., M.S., and V.S. reviewed and edited the draft.

## Funding

## Competing interests

The authors declare no competing interests.

## Additional information

[1]Potsdam Institute for Climate Impact Research, Potsdam, Germany. [2]Helmholtz Centre for Environmental Research (UFZ), Leipzig, Germany. [3]NSW Department of Primary Industries and Regional Development / University of New England, Armidale, NSW, Australia. [4]Geography Department, Humboldt-Universität zu Berlin, Berlin, Germany. [5]International Institute for Applied Systems Analysis (IIASA), Laxenburg, Austria. [6]Climate Analytics, Berlin, Germany. [7]Center for Global Sustainability, University of Maryland, College Park, USA. [8]Institute for Geology, University Hamburg, Hamburg, Germany. [9]Department of Applied Ecology, Hochschule Geisenheim University, Geisenheim, Germany. [10]GEOMAR Helmholtz Centre for Ocean Research Kiel, Kiel, Germany. [11]Priestley International Centre for Climate, School of Earth and Environment, University of Leeds, Leeds, UK. [12]Centre for Environmental Policy, Imperial College London, London, UK. [13]La Follette School of Public Affairs, University of Wisconsin-Madison, Madison, USA. [14]Net Zero Lab, Max Planck Institute for Innovation and Competition, Munich, Germany. [15]Centre for Energy, Environment, and Natural Resource Governance, University of Cambridge, Cambridge, UK. [16]Group for Sustainability and Technology, ETH Zurich, Zurich, Switzerland. [17]Research Centre for Carbon Solutions, Heriot-Watt University, Edinburgh, UK. [18]Evidence for Policy & Practice Information Centre, Institute of Education, University College London, London, UK. [19]Kiel Institute for the World Economy, Kiel, Germany. [20]Institute of Biological & Environmental Sciences, University of Aberdeen, Aberdeen, UK. [21]Smith School of Enterprise and the Environment, University of Oxford, Oxford, UK. [22]Florida International University, Department of Earth and Environment and Institute of Environment, Miami, FL, USA. [23]Global CO2 Initiative, Department of Mechanical Engineering, University of Michigan, Ann Arbor, MI, USA. ✉e-mail: sarah.lueck@pik-potsdam.de; jan.minx@pik-potsdam.de

