## [Transparent Peer Review file · Nature Communications]

Scientific literature on carbon dioxide removal revealed as much larger through AI-enhanced systematic mapping

Corresponding Author: Dr Sarah Lück

Version 0:

Reviewer comments:

Reviewer #1

(Remarks to the Author)

This paper contributes a novel machine learning-assisted systematic mapping methodology to provide a comprehensive evidence map of the CDR literature. The manuscript is well-written and motivated. The written language is simple and easy to be read for people of a different background. The novelty of the study is also presented clearly and the systematic mapping approach is comprehensively presented in the supporting documents. Overall, a very useful and very much needed tool. With some additional filters this tool has the potential to provide a great service to the CDR community and climate change mitigation efforts in the wider context.

However, I miss a reflection on the potential reason for certain results, which would be helpful for further guidance. Also, the discussion could benefit from clear recommendations for the CDR community on where to focus on (and why?).

Also, it would be great to see a progression of the tool to more specificity (which might have been outside the scope of this work). For example:

- For BECCS, further coding based on biomass origin, for DACCS, further coding based on type of sorbent/solvent used, etc. This would allow an overview of the CDR literature space not only based on macro-CDR methods, but going more into the specific technology (e.g., high-temperature vs. low-temperature DACCS).
- For LCA studies categorize the function of the system (heat generation, biofuels production, application to soil, CO₂ sequestration), which would help in refining the literature landscape overview.

Main comments:

- "In policy, CDR has gained increasing attention in recent years (Geden, Scott, and Palmer 2018; Schenuit et al. 2021), but many countries still lack concrete policies to scale CDR (Smith, S. M. et al. 2023) Chapter 5). This has led to a considerable gap between countries' (so far limited) plans to develop and deploy CDR versus CDR's required role in mitigation scenarios that stabilise global temperatures at an increase well below 2°C. (Bellamy 2018; Cox and Edwards 2019; Schenuit et al. 2021; Smith, S. M. et al. 2023). (page 2, main)

o I would specify that the quantitative information of CDR required in mitigation scenarios is obtained from model-based scenario analyses. Hence it is a number resulting from modeling assumption and model "fingerprints." I would find it important to specify that there is still a large spread of uncertainty associated with the scenario output: 1) Structural uncertainty (regarding technological innovation and uptake, market behaviour, etc. Assumptions in the models), 2) Parametric uncertainties (involving differences in parameter calibrations, regional and temporal scale) and 3) Fundamental modelling choices (e.g., optimization versus simulation frameworks), model structure and foresight. (Dekker et al., 2023, Nature Energy)

- In Figure 3, it would be interesting to see, for the location-based research the correlations between CDR methods and the specific geographic locations studied. Is there a difference between this map and the results of Fig. 3a-3b?
- In long-term mitigation scenarios that achieve the Paris long-term temperature goals (Byers et al. 2022) mainly BECCS (99%), afforestation (67%) and DACCS (29%) are the CDR options included. There is not a single scenario dealing with biochar or soil carbon sequestration, although these methods can deliver co-benefits such as food security or N₂O emission reduction (Lehmann et al. 2021).

o Can we contextualize this statement more? Do the models not include biochar and soil carbon sequestration in their representation (i.e., the reason why these methods do not appear in the scenarios), or are they represented, but the models do not choose these methods in mitigation pathways (if so, why)?

- Coding Framework: Long-term storage is defined as at least 5 years. As 5-years is not a climate-relevant time horizon, why

wasn't a longer time horizon chosen in the definition of long-term storage when screening for CDR literature?

- Figure 1 should be better explained. For example, the figure is not clearly explained in the text or caption (I know the methods are at the end in NC but this is not easy to follow and read to have this figure here). For example, it is unclear why and how documents are coded by hand. Why is each document coded by two people, does this mean all 30,000 papers?
- The authors report the results consistently, however, I would be interested for the reason of reported results, e.g. why some CDR methods are more popular in certain regions. I would suggest to elaborate on this in either the results or the discussion. For example;
'China focuses more on biochar research, Europe on BECCS, and North America, particularly the US, on DACCS relative to the research shares per technology in all countries. Research on ocean-based CDR methods is more predominant in Oceania and North America'
Can the authors give reasons why (in China)? For example, certain regimes, policies implemented, social pressure/acceptance? This would be useful to better understand how certain aspects can impact research efforts.
- I also do think that the results can be better structured in terms of paragraphs. Please think about the key messages you want to provide. For example, in section 2.2, the authors are going a bit back and forth between the number of studies of CDR options while I don't think that for each method one has to provide the results using its relative share of overall number of studies.
- If possible, please provide the search terms in the methods or SI.
- Maybe I missed it, but why is storing CO₂ in wooden buildings or concrete not considered as CDR option?
- On the one hand, I liked the benchmarking of the research landscape with regard to the IPCC report. On the other hand, I am not sure how useful it is, i.e. I am not sure whether the IPCC should reflect the overall underlying scientific literature or instead should just choose the most relevant studies.
- The discussion/conclusions seem more like a repetition of the results. They should be partly revised by provided a set of recommendations for the CDR community.
- It would be great if you mention somewhere in the paper, what makes this tool so useful and be specific about it. Highlight the added value of using machine learning but also the novelty aspect of your work.

About the tool: If possible, it would be great to have some additional filters in the tool itself;

1. Filtering the papers by research gap would make a major difference and position this tool above any existing online tool used for literature mapping as of today. It would probably mean a lot of work but think about the functionality of your tool. This would in turn increase its usability and expose your work widely in the research community and beyond. In addition to this, it would also be very interesting to have a discussion in the paper around what are the major research gaps being addressed as of today.
2. In the LCA studies, one of the most important elements when putting together the study is the functional unit of the assessment which connects to the function of the system being analysed. For example, if I am looking at an LCA study of biochar, the system at hand could focus on the environmental impacts of biochar production. Other functions biochar is used for concerns: a) energy production, b) biofuel production c) biochar application to soil for carbon removal d) biochar used for soil purification etc. My point is, it is of crucial importance that you add another layer of filter where the LCA studies are filtered by the function of the system. For example, as of now the tool results in roughly 200 LCA studies on biochar but if I want to look only the studies on biochar application to soil, I will not get more than 6 to 10 LCAs. This would be a major innovative element of your tool.
3. In the same line as the LCA part, it is very important to provide a filter that specifies the IAMs scenarios used for the studies that work with them. And this is another point that can be discussed in your paper.
4. Add another filter concerning the Technological readiness level (TRL) of the CDRs. In a way this could as well explain why there is a certain focus on specific CDRs compared to some others. In terms of functionality this would be very useful for different stakeholders beyond research (investors, governments) who want to learn mainly about the technologies that are ready for large scale implementation.

Other minor comments:

- Fig. 3 caption: "right" and "left" panel are inverted in the caption text
- Data and Methods section, page 15, first paragraph: SI-Figure 5 instead of SI-Figure 4
- Supplementary figures and numbered incorrectly:
 - o E.g., SI Figure 1, related to Figure 3; instead of Figure 2
- Can we not use machine learning in the title instead of using artificial intelligence, in the methods machine learning is used while in the title AI (I am aware that machine learning is a subset of AI though).
- I would refer to the online tool available on the web, somewhere in the paper. Is it open source and maintained?
- I would suggest removing the last paragraph on page 4, the one that summarizes the findings in the introduction section. It is a great paragraph, but you mention these findings in the abstract, discussion and conclusion. I think it would help to reduce any sense of repetitiveness.

Reviewer #2

(Remarks to the Author)

Reviewer #3

(Remarks to the Author)

Reviewer #4

(Remarks to the Author)

Carbon dioxide removal (CDR) and negative emissions technologies are needed in all scenarios that achieve global warming targets set by the Paris Agreement, and therefore are a crucial consideration for policy makers. This article utilizes emerging applications of artificial intelligence models to generate an automated systematic map of available academic evidence relevant for CDR. This allows the authors to characterize the distribution and extent of academic evidence across relevant metadata variables such as publication year, first author affiliation, intervention type, methodology and scientific discipline. In my view, an important and key contribution of this research is discrepancy that was revealed between the distribution of academic evidence, policy and deployments across different CDR technologies. This finding is particularly timely over the coming decade, as there is a strong motivation to scale up CDR (particularly in novel technologies). This systematic map provides valuable insight into knowledge gaps that can inform future research priorities, as well as under-exploited knowledge clusters that can inform CDR investment and deployment.

I want to congratulate the authors on this work, which represents a valuable application of how machine learning tools can be harnessed to conduct evidence synthesis with a large enough scope that is relevant for policy makers. I recommend that this article be accepted for publication, with revision. I have made most of my comments in the attached annotated pdf file, however include more general comments below. In particular, I think the discussion should elaborate more on key findings – specifically how patterns of CDR research differ compared to policy and practice, as well as providing more information and clarity in the methods with respect to the named entity recognition. For all the figures, please verify that the color palettes used are colorblind friendly where possible.

Introduction

- Many valuable points are presented in the introduction, however I think a few contours can be better defined that would help to position this work, its motivation and contribution – specifically with how increasing scientific literature interacts with trade offs between scope and available time resources when conducting systematic evidence syntheses, and how automated approaches can be applied in this context.
- Many different types of interventions are discussed, and the inclusion of a typology figure or table that provides a short description of the main classes could be helpful to clarify the scope of interventions assessed in this map (essentially a summary of the table in the Coding Guideline – Categories – 0 Technology (CDR Technologies). This would also clarify the boundaries between the different types used in Figure 2 – e.g. blue carbon management can be a form/sub-type of soil carbon sequestration? I would suggest including a figure like this and moving table 2 to the supplement

Results

- More precision/evidence could be added to several statements (e.g. briefly citing F1 statistics, confidence intervals, etc where relevant)
- Section 2.1 & Figure 2: It seems for all the methods, that annual growth rate starts high and then declines, perhaps because it is easier to grow by a larger percentage when absolute numbers are low. So I wonder how useful a direct comparison is between growth rate for all publications in climate research vs CDR when growth rate seems so affected by the absolute size of the literature base? The overall size of the CDR literature base is given, but what is the size for all publications on climate change, and based on their differences in size, is a direct comparison in annual growth rates suitable?

Discussion

- See comments in main text file, particularly my recommendation to elaborate further on the disconnect between policy and practice and scientific knowledge

Methods

- A note on data and code availability – will the raw data on the articles included in the study, and their predicted relevance for the different metadata variables, be provided along with source code (e.g. in an online repository)? This would be a valuable resource to the CDR community.
- I did not see detailed in the methods how the named entity recognition in titles and abstracts was performed – could you please specify?

Reviewer #5

(Remarks to the Author)

Review report

Submitted article: Scientific literature on carbon dioxide removal much larger than previously suggested: insights from an AI-enhanced systematic map

Date of review: 3/7/2024

Synopsis

The authors predict a much larger literature on CDR than previously found. They use a state-of-the-art supervised machine learning approach, but I am unable to judge if this is a methodical development, or just an advanced, but common, method applied to a new topic, namely CDR. The two central findings of the paper are the large size of the CDR literature and the large focus on biochar. There are four findings of secondary importance: the affiliations of reviewed first authors are concentrated in China and OECD countries; most of the literature is categorized as technology-studies; the reviewed methods are mainly experimental or modelling-based; and the distribution of CDR types in the literature is different from that cited by IPCC and that in policy.

Significance

The application of machine learning is not a major scientific contribution on its own – and nor is it framed as such by the authors. If it was the main contribution, a method-oriented journal may be more fitting. When ML is used in a complementary fashion it can yield valuable insights, but the submitted version is mainly descriptive, and is only an incremental improvement to previous published reviews on CDR. While the authors prediction about the size and the distribution of the CDR literature are important and interesting findings, these are not put into play sufficiently to give further insights. The finding that biochar dominates the literature is interesting, but no empirical effort and almost no theorizing to understand what could explain this dominance. While the two main findings are important and of general interest, I believe they are incremental and that the paper in its current form is lacking an investigation into the results.

Recommendation

Behind the paper is a substantial effort to screen and annotate a training dataset. The methods are rigorous. The explanation of the machine-learning approach is also easy to follow. The two central findings are of general interest.

There are two main reasons I recommend a rejection of the paper in its current form.

First, the major issue with clarity renders many arguments in the paper inaccessible. Many basic mistakes (referencing and formatting) coupled with unclear writing and multiple off-hand claims give the impression of insufficient internal editing. While I strive to separate form from content when giving critique, the lack of clarity detracts significantly from the overall quality of the paper. Second, while the two central findings are interesting, there is no investigation into potential underlying explanations. The attempts to contextualize the results are weak and occasionally based on a flawed premise. For instance, the two claims - that citation patterns in IPCC should mirror the distribution of CDR types in the predicted literature and that the distribution of literature point to evidence gaps on certain CDR types – are not convincingly justified. The secondary findings are of limited general interest, and without deeper analysis the central findings remain primarily descriptive.

I encourage the authors to revise the paper thoroughly and resubmit to another journal, since the work contains important results that would be of general interest to the research community, and potentially, as the authors also mention, to the coming work of the IPCC cycle.

Note: While I am confident to provide intermediate judgment, I am unable to provide an expert judgment of the machine learning approach. I have some, but not much, practical experience with applying supervised machine learning and transformer-based language models. From my judgement, the authors' methods are both rigorous and based in solid experience.

Main points

The main argument of the paper is that the CDR literature is much larger than previously thought. This claim could be strengthened by testing whether the algorithm is overly inclusive i.e. predicts too many false positives. I understand the precision of the algorithm is measured on the training data, but further robustness could be checked by sampling some predicted documents. As of now the authors only test that the predicted documents are sufficiently inclusive (validation against a list of IPCC references). There might be advanced ways to do this, but it could also be as simple as extracting a random sample of documents and manually screening how many are relevant.

Figure 1 is well-designed and succeeds in conveying the machine learning approach.

A general issue is the vagueness of what constitutes a CDR study, and the subcategories. For instance, what does it mean that a document is classified as biochar research. The specific in-/exclusion criteria are found in the SI, but a general explanation in the main manuscript would be useful. The many ways in which classification is described leaves the reader unsure "Biochar research is covered in [x] publications"; "Roughly one third of CDR research ... refers to ..." "there is 3-4 times more research related to CDR" (this sound as if the literature is not mainly on CDR, but something else that is related); "CDR research addresses"; "the second largest category, SCS, with 23% of the total literature"; "BECCS represents only about 5% of all studies" (Side note: BECCS doesn't represent any studies, it might be represented in some studies though); "the literature on CDR in general features mainly policy". These examples raise questions of what it means that a study is on a type of CDR such as whether tangential attention sufficient for inclusion, is a mention in the title, abstract and keyword sufficient? What about an IAM-based study that includes, but is not focused on CDR? How many studies contain multiple CDR types? I don't need answers to these specific questions here, but it would be a great improvement if such questions don't arise when reading the submission. I also suggest adding a clear definition of what is meant with a CDR study in the methods section, if not in the main. Of course, this can only be a general definition, that goes for all CDR types.

The amount of predicted CDR literature is compared with previous scientometric studies, but it is unclear if the comparison accounts for the different time of the studies. Further, several of the authors have been part of previous reviews of CDR but found a lot less literature (Minx et al. 2018, Minx et al. 2017) in their search i.e. before screening. Since AI is only used to screen and not to search, it would be of general interest to explain why the initial search (unscreened) results in ~70,000 articles compared to ~6,200 in the 2018 study and 2900 in the 2017 study. Does this review cover more CDR types or were important keywords missing from the previous search queries? How much of the difference is explained by literature published after the previous studies?

The authors emphasize multiple times that the literature cited by IPCC doesn't reflect the predicted literature: "IPCC assessments are not a broad reflection of attention patterns in the underlying scientific literature"; "Only 1% of the available scientific evidence on CDR was cited in the IPCC AR6", and "IPCC assessments cite a much lower share of reviews on CDR relative to the overall literature" (which is followed by the harsh and unfounded claim that this means that "IPCC therefore also fail to indirectly assess the available literature"). This emphasis is repeated for biochar, which dominates the predicted CDR literature, whereas IPCC is dominated by BECCS. However, it is not convincingly explained why IPCC citations should reflect the distribution of studies on each CDR types. In the same vein they write "understanding which subjects [CDR types?] receive more or less attention is vital" – This can be understood simply by looking at the IPCC references study i.e. for this, their mapping is unnecessary. Continuing: "to ensure that IPCC reports continue to provide a balanced assessment of the evidence". A balanced assessment of evidence connotes a pro-et-contra, but since this make little sense in this context, the reviewer is left to guess what the authors mean; should it be ensured that IPCC continues to cite CDR type proportionally to their distribution in the literature? (Side note: why say continue if IPCC doesn't reflect it already, as was just claimed). I cannot see why such proportionality is important, but no arguments are given anyway. The authors seem to hold an underlying belief that x number of studies on CDR type A has a weight comparable to an equal number of studies on CDR type B – and therefore that the ratio (or relative number) of studies can indicate where more research is needed (e.g. "can help identify topical areas worthy of focus"; "we find substantial evidence gaps"). I am sceptical to the number of studies can be used in the way done by the authors, but even if so, it is a very noisy signal, and interpretation should therefore be cautious. This critique above also applies to section 2.6, i.e. it is unclear why it is relevant or of value to compare how studies are distributed across CDR types with "other indicators of attention". If this critique misses the mark in the way that the authors are not claiming that attention patterns should match, then it is still unmotivated why the authors want to compare patterns.

The article compares the growth of CDR research to the growth of climate change literature, but it is not motivated why this is an interesting comparison, and I am unsure if it is a valid one. Research fields have highly different traditions of publication, e.g. because the cost and scale of experiments differ. I'm sure better ideas exist, but for a start a comparison with the mitigation literature or total scientific output across all disciplines would be improvements.

The three abovementioned critiques can also be related to David Humes famous "is-ought" problem. This issue was acknowledged in a previous publication by two of the co-authors (Callaghan, Minx & Forster 2020). This issue applies equally to this paper, although there is no acknowledgement of the issue, in this or in other terms.

The authors make an interesting note in mentioning that high biochar publication rates might be due to its potential co-benefits. Such remarks bring the authors knowledge about CDR to the front and is more relevant to a general journal such as Nature Communications than advanced scientometric methods.

In paragraph 8, it is argued that by using AI, the authors can assess the "balance of research" (unclear meaning) in terms of CDR types, research methods etc. But this can also be and is often done without AI. With 26 authors and 7 assistants in the acknowledgement, each would have to screen less than 4,500 articles (by titles first and abstracts only where necessary) and annotate less than 1600 documents (double-coding included). These numbers (70,000 and 29,000 divided by the number of co-authors and assistants) can be greatly reduced by first screening by journal subject areas, journal titles, and for irrelevant keywords in journal and document titles. This is not to say that there isn't a valid motivation for AI, but rather the one stated is insufficient.

Generally the treatment of CDR is strangely uncritical although CDR is highly contested and politicized (see e.g. Anderson et al. 2023; Carton et al. 2023). Several of the co-authors have already contributed to this debate. To be more specific: The abstract and introduction emphasizes CDR as a critical strategy to limit global warming below 2°C. First, if any strategy is critical, it is emission reductions, whereas CDR is a supplementary strategy (Zickfeld et al 2023). CDR is meaningless if not combined or preceded by emission reductions (Fankhauser et al 2022). The authors mention three roles of CDR. The first role warrants further qualification, since even for residual emissions (hard-to-abate sectors), emission reductions are a preferable strategy to CDR in terms of several indicators (Fuhrman et al 2024). The second role hinges on a misguided fixation on the 2°C goal, rather than the objective of limiting global warming as much as possible. In other words, if emission reductions do not occur rapidly, reductions (and not CDR) will only become more urgent to minimize further global warming – which should be the target of climate change mitigation, rather than expecting CDR can be scaled to match current (i.e. unmitigated) emission levels. This fixation also ignores the expected (and those unknown) detrimental effects of temperature overshoot, even if temporary. The third role of CDR is indeed important, as some CDR options (land-based are mentioned) can lower net emissions and already does so (as mentioned by some of the co-authors in other publications). But any land-based CDR is limited by exactly that – land. In addition, climate change is not the only ecological crisis, which means there are other challenges to large-scale deployment of CDR beside the limited potential for removals. Many CDR options are found to have negative environmental effects (see e.g. Heck et al, 2018; Prütz et al, 2024; Wood-Hansen & van den Bergh 2024).

Specific minor points

The authors mention they find little research on how to build CDR economies where the CDR (...) provides economic revenues beside mitigating global heating. This is a very specific example; an example of such literature would be interesting. It is however unmotivated why CDR should provide economic revenue beside mitigation – this seems a problematic requirement.

What is meant with a diverse range of interdisciplinary fields and why does it add to the complexity of reviewing literature? I

would think interdisciplinarity would enable reviews, in contrast to unconnected fields.

It is mentioned that “there is little CDR research on the societal implications [of CDR] such as public acceptance, governance”. The choice of words here is imprecise and unfounded. The mention of societal implications is unclear – nowhere else are these mentioned. The reader is left to speculate if societal implications are understood as a subset of studies predicted to fall in the “context”-classification as “Policy / Government” or “Public perception”, but none of these equal “societal implications”. Note also that public acceptance specifically refers to public perception of policy after implementation (as opposed to acceptability which is *ex ante* policy adoption).

The use of the Gini-coefficient to explain the distribution of the literature is unnecessary, as there is no reason to expect an equal distribution.

Section 2.4 refers to Keller et al. 2018 and Werner et al. 2022, but it is unclear what the role of these references is. Are they examples of studies under the label “policy and governance”, or are they meant to support a claim? If the former, please help the reader, for instance by writing, “see e.g.”. The same sentence claims that “policy and governance” and “earth system” are important categories but doesn’t explain why.

The authors find that the reviewed BECCS studies mainly fall into the “modelling” category and infer that this “points toward limited efforts to research the design of the technology itself”. This inference is invalid, why would an increase of modelling studies using BECCS imply anything about other research efforts into BECCS?

It is claimed that “There is not a single scenario dealing with biochar or ...”, but where does this claim from? Is it a result of the review?

p. 13-14 states that the ML-assisted approach enables the authors to be systematic in their procedure, but as they also mention in the introduction there are also systematic reviews not relying on ML.

It is stated that DACCS is a place-specific CDR-type, but as I understand it, DACCS might be one of the least context-dependent types of CDR, except for the importance of local access to renewable energy? In the same paragraph is the trivial observation that there are fewer studies on novel technologies.

The authors call for more humanities and social science research to support evidence-based decision making. While I have sympathy for the point, the argument needs sharpening. First, the authors should be aware of the critique of the notion of evidence-based policymaking (see e.g. Head 2010). Second, the humanities do not contribute to evidence-based policymaking, nor is it the ambition.

The authors suggest using their literature map to identify what has been “under-cited” by the IPCC. This is an interesting idea and sounds potentially useful. It could be an interesting addition to the paper. If not, please elaborate how this idea could be carried out.

The authors make an unbacked and imprecise claim about what drives political attention (i.e. that relatively slow mitigation action will shift political attention toward CDR). What political attention - the media, the public, or that of political elites? It is entirely speculative that there should be a causal relation between a lack of emission reductions and an increasing attention toward CDR. If we are to speculate, the short-term prospects of economic growth and job creation through technology-exports associated with investments in CDR combined with the hope (among vested interests) that CDR can enable continued fossil fuel extraction are more likely drivers of elite political attention. How close we are to the 2°C goal is probably the least crucial factor in where political attention goes. The political economy of climate change is tightly associated with hegemonies and dominant processes of power i.e. capital accumulation.

Please provide colourblind-safe versions for the figures in which colour is the only way to distinguish elements. These can be put in the supplementary material. (See the Springer Nature commitment to inclusion)

On methods

The main manuscript is lacking cut-off date for the search queries. The only place I can find it is in a figure in the supplementary material, but it is only given by the month.

In section 3 “our classification system [...] is not only able to predict the relevance”. Does it score of the degree of relevance of each study, or is it a binary sorting into relevant/irrelevant.

Section 2.4 mentions the use of “journal information” without further explanation.

The difference between the annotation mentioned in paragraph 3 (screen and annotate) and 4 (we further annotate) is unclear.

The authors claim to achieve high levels of recall and precision. These terms are not my expertise, but I understand recall to be a ratio, but it seems here to be used in the sense of resulting in many search results. A footnote or explanation could be included in the supplementary material. The same goes for precision; it is unclear how it is measured. I understand these may be related to the classifiers and their scores such as F1 and ROC-AUC. Even though this is technical jargon, it can be explained in simple terms in the supplementary material.

The authors acknowledge that their classification system is not perfect, which is a trivial point. It is then argued that manually compiled evidence maps are assumed to be perfect, which sounds like an unnecessary strawman.

On form and style

The lack of basic formatting in the supplementary materials should not need mentioning. Please provide page numbers in all documents. The lack of formatting in the document headed “Supplement” is not helping the reader (title, section headers, and subsection headers). All tables in the other supplementary material lack numbering and titles. For ease of reference, please add numbering the supplementary materials (e.g. SI 1, SI 2).

The reference style is inconsistent in multiple ways, both in-text references and in the reference section. Examples can be found right next to each other, see for instance the top of p.3: (Jan C Minx et al. 2018; J.C. Minx, Lamb et al. 2017). The authors may have intended to clean such formalities up after receiving a journal decision, but this is inconsiderate to the reviewer.

The text is lacking clarity, which makes it hard to follow the intention of the authors in multiple places. The text needs a major edit of its writing. Some mistakes are basic, leaving the impression that the final text was not read from A-Z before submission. Some examples:

- Repeated sentences, one of which is used three times (including the references) in addition to the mention abstract. E.g.

“CDR is a critical component of any strategy.”, “the CDR literature is 3-4 times larger than previously estimated (Burns 2021; J.C. Minx, Lamb, et al. 2017).”

- Smith, S. M. et al. 2023 is referenced six times in the first four paragraphs. Another article (of the last author) is mentioned 8 times. Repeating a reference can be warranted, but not to the degree shown here.
- Figures are placed before they are introduced in the text.
- Unclear, unnecessary, mistaken and repetitive use of words. Some of these can be guessed by a specialist, but this should not be necessary. [Note that my formatting seem to have disappeared in the manuscript tracking system from the below list of examples. The formatting was meant to indicate a line or word copy-pasted from the submitted article and my comment in parentheses]
- e.g: important prerequisite (pleonasm); new scientific developments (tautology); actively remove CO2 (what is meant by active?) sustainably(?) scaling CDR; sustainably achieving gigatonne-scale; grows at ever(?) increasing rates; map out; methodologies vs. methods; “while research on other CDR methods” (are “other” those which are not land-based as mentioned earlier i.e. ocean-based?); Other biological CDR methods (the classification into biological CDR methods is not introduced previously); engineered CDR options (again, this is not an established category of CDR); growth rate of BECCS is... (BECCS doesn't grow, the number of studies grow); Place-based research is important for ... in-situ; efficacy (effectiveness?), efficiency (what type?); provide an assessment at scale (“at scale” does not make sense); Most studies focus on investigating the CDR method from a technical perspective where the method itself is investigated; topical areas (are these simply topics?); distinctly different (unclear meaning); while patenting activity has been active in BECCS (most active or only active?); the literature is dominated by biochar research today (today -among articles published in 2024, or in the accumulated literature as indicated elsewhere?); with a geographical centre in China (institutional affiliations with China or studies in Chinese contexts?); most recent 6th Assessment Report (there is only one AR6); On p. 19, first section: research designs vary substantially across CDR methods (...) across individual CDR options research designs vary substantially (These sentences are in the same section, unclear if they mean something different?); please be clear on the use of compare with and compare to; we find further a strong focus on research on the methods (I assume the CDR literature focuses on methods, not research on methods); relative to the research shares per technology (this wording occurs several places and is very hard to understand); one third of CDR research in the literature (redundant); more place-specific research is mentioned in the same sentence as more specific sub-national locations (unclear if these terms mean the same); compared to vs. compared with.
- Section 2.4 mentions “Our framework”, but no framework is mentioned elsewhere.
- In methods section: ClimateBERT is chosen here due to its better performance (see Supplementary Information). Please be more specific when there are multiple supplementary materials with multiple sections.
- In methods: summarized in Figure 1 and SI-Figure 4. The authors must be referring to SI-Figure 5.
- In the note to Figure 4: Please improve the descriptions of the panels, they are currently difficult to understand.
- There is no Table 1, and Table 2 is never introduced in the text.
- The panel descriptions of figure 3 doesn't match the panel positions.
- When referencing a panel in a composite figure, it helps the reader if the reference specifies the panel (e.g. see Figure 3A).
- What is meant by support in “a stronger focus on scenarios is supported by the fact that we find a shift from experimental research to modelling work”.
- Clarify when CDR in general refers to CDR in general and when it refers to a specific category.
- The union of “Social science and economics” is the same as saying economics, since economics is a social science already. Do the authors mean social science or economics?
- In section 3: The sentence “our systematic map is scalable to the entire research domain around CDR” is confusing, as it sounds like the map isn't scaled yet.
- The authors claim they perform the most comprehensive effort yet, but I assume something else than effort is meant.
- In paragraph 3 it seems the authors aim to introduce a list of knowledge domains. If so, please introduce the list properly and clarify when one list element ends, and another begins.

Paragraphs 5-7 of the Introduction seem to partly overlap. It is also unclear why paragraphs 3 and 6 are separate (e.g. “a number of studies has attempted to synthesize these growing literatures [references]” and “the few available overviews of the field have become outdated [other references]”). The paragraph structure is unclear and confuses the reader. I suggest either a restructuring or making the structure of existing paragraphs clearer.

The abstract would benefit from more specificity. For instance, why not mention biochar instead of “very concentrated on specific CDR options”. The use of “policy and practice” as it stands in the abstract.

In Figure 2, it is unclear if the middle panel shows accumulated shares, or the shares for each year of publication. It is hard to see the information presented in the bottom panel. Could it be enlarged by moving the top figure (less interesting) to the appendix, alternatively splitting the bottom figure into two; one representing the two lines “all publications on...” and another with the CDR types?

Figure 4, top panel. The axis label “publications mentioning technology” is confusing, especially given the text right before mentioned “technology research” as a category. If you mean “CDR method” (as used elsewhere), please be consistent.

Middle-panel: Many journals, such as Nature Communications, are interdisciplinary journals. Such a “research field” seem to be missing from the diagram.

The types of CDR are sometimes referred to as CDR methods – this causes a tiny moment of confusion, as it could refer to the methods used in the CDR literature. I suggest avoiding the use of CDR methods and stick to other terms e.g. options, types, technologies, techniques.

On the Supplementary material “Instructions for Coding”

CDR-Syntra is described as a living database, but the use, purpose and accessibility of this database is not mentioned elsewhere. The “living”-feature is unclear to me, despite the elaboration (p1, list element 3); wouldn't an automatically updated test dataset also require retraining the algorithm, as to ensure the training data is a valid sample of the complete

dataset?

The readability of the many tables could be greatly improved with simple formatting, mainly to ensure that the tables are broken across pages as little as possible. Even if the authors planned to do such final formatting after the peer-review, it is considerate to the reviewers if such formatting is done before submission.

p. 19: The section “assessment” introduces a categorization of documents, which is not used in the manuscript. It would be an interesting addition, which could provide some of the lacking insight of what the large number of predicted studies focus on (and potentially into the constituents of the “technology” label).

The “technology” label is very general, if not outright unclear. This is also evident from the large share of documents with a “main focus” predicted as “technology” (89 %). This warrants a decomposition of the label, otherwise it contains little informational value. The description of the label is described in the “main focus”-table (in “Instructions for coding”) as “the most vague/catch-all label”, which is problematic since it suggests that coders assign difficult-to-classify documents to this category. This almost sounds as if the authors were already aware that this label could be improved from a re-naming or decomposition. Please do so.

On the “Supplement”-file

A table of contents would be helpful.

SI Figure 5: I suggest adding the number of manually screened and annotated documents to the figure, as to give the complete overview of the procedure.

The number of “documents after deduplication” differ in the main manuscript and in SI figure 5 (75,518 vs. 69,942).

Please give a brief introduction to the meaning of the indicators F1 and ROC and the the scoring scale (e.g. 0 to 1, where 1 means...).

References (only for those not already in the article)

Anderson, K., Buck, H. J., Fuhr, L., Geden, O., Peters, G. P., & Tamme, E. (2023). Controversies of carbon dioxide removal. *Nat. Rev. Earth Environ.*, 4, 808–814. doi: 10.1038/s43017-023-00493-y

Brian W Head, Reconsidering evidence-based policy: Key issues and challenges, *Policy and Society*, Volume 29, Issue 2, May 2010, Pages 77–94, doi: 10.1016/j.polsoc.2010.03.001

Carton, W., Hougaard, I.-M., Markusson, N., & Lund, J. F. (2023). Is carbon removal delaying emission reductions? *WIREs Clim. Change*, 14(4), e826. doi: 10.1002/wcc.826

Fuhrman, J., Speizer, S., O'Rourke, P., Peters, G. P., McJeon, H., Monteith, S., ...Wang, F. M. (2024). Ambitious efforts on residual emissions can reduce CO2 removal and lower peak temperatures in a net-zero future. *Environ. Res. Lett.*, 19(6), 064012. doi: 10.1088/1748-9326/ad456d

Prütz, R., Fuss, S., Lück, S., Stephan, L., & Rogelj, J. (2024). A taxonomy to map evidence on the co-benefits, challenges, and limits of carbon dioxide removal. *Commun. Earth Environ.*, 5(197), 1–11. doi: 10.1038/s43247-024-01365-z

Fankhauser, S., Smith, S. M., Allen, M., Axelsson, K., Hale, T., Hepburn, C., ...Wetzer, T. (2022). The meaning of net zero and how to get it right. *Nat. Clim. Change*, 12, 15–21. doi: 10.1038/s41558-021-01245-w

Heck, V., Gerten, D., Lucht, W., & Popp, A. (2018). Biomass-based negative emissions difficult to reconcile with planetary boundaries. *Nat. Clim. Change*, 8, 151–155. doi: 10.1038/s41558-017-0064-y

Wood Hansen, O., & van den Bergh, J. (2024). Environmental problem shifting from climate change mitigation: A mapping review. *PNAS Nexus*, 3(1), pgad448. doi: 10.1093/pnasnexus/pgad448

Zickfeld, K., Maclsaac, A. J., Canadell, J. G., Fuss, S., Jackson, R. B., Jones, C. D., ...Zaehle, S. (2023). Net-zero approaches must consider Earth system impacts to achieve climate goals. *Nat. Clim. Change*, 13, 1298–1305. doi: 10.1038/s41558-023-01862-7

Version 1:

Reviewer comments:

Reviewer #1

(Remarks to the Author)

Thank you for making the revisions based on the comprehensive set of comments. Although some of the comments cannot be entirely satisfied for various reasons, I believe the paper has been substantially improved. I would advise to accept the current version of the manuscript.

(Remarks on code availability)

Reviewer #2

(Remarks to the Author)

(Remarks on code availability)

Reviewer #3

(Remarks to the Author)

(Remarks on code availability)

Reviewer #4

(Remarks to the Author)

I thank the authors for their diligent responses to my revisions and comments, and support the revised manuscript for publication.

The additional elaboration on the role in advancing the frontier of AI-assisted systematic maps, as well as our understanding of the field of CDR better highlight the novelty and contribution of this work, and the addition of the new Figure 2 emphasizes the large perimeter of this synthesis.

I feel that the revised document presents a clear explanation of the results, and well-supported interpretation and evaluation of their relevance. The authors made valuable additions for interpreting the accuracy and precision of their findings by including the performance and confidence intervals of the classifiers, as well as included numerous statements that better define the limitations of interpreting certain patterns and trends (e.g. high growth rates at low publication numbers in Fig 3). The revised section "2.5 IPCC reports, policy-making and practice have foci different from the scientific literature" enriches the comparison between the distribution of academic literature and publications informing policy. With regards to methodology, my previous request to add clarity with regards to the process for named entity recognition has been addressed.

(Remarks on code availability)

Reviewer #5

(Remarks to the Author)

The authors have thoroughly revised the manuscript, and I now recommend acceptance given some minor revisions.

General Comments

The authors have addressed most of my initial comments. They have rewritten entire sections, improved the clarity of their writing, removed hyperbolic language, introduced Carbon Dioxide Removal (CDR) in a more balanced way, and provided stronger reasoning for their results. Collectively, these changes make the article more accessible and nuanced in its findings.

Several new and interesting paragraphs have been added to the discussion. However, there remains some repetition from the results and introduction, offering an opportunity to shorten the text for improved readability.

Feedback on the reply to reviewer

I found the tone of the reply document less than collegial and in places, replies appeared directed more toward the editor than the reviewer. This impression stems from several observations:

1. The authors repeatedly refer to other reviewers, which is irrelevant to my review process, as I had no access to their feedback. Replying to a comment is more convincing than referring to the judgment of another reviewer.
2. Phrases like "_this part has been deleted on request by reviewers 1 and 4_" appear twice in reply to my comments. Acknowledging reviewer 5's feedback equally would have been appropriate.
3. Statements such as "_the decision regarding whether we should submit elsewhere ultimately lies with the editor_" are unnecessary and dismissive. This is self-evident, but reviewers are expected to make recommendations to the editor; I specifically used the term "encourage."

The authors seem more annoyed than appreciative of the peer-review process. This is disheartening given the voluntary nature of reviewing and the detailed feedback provided in my first review, which included examples and actionable corrections.

The authors note that they "_generally do not agree with this harsh judgment_" regarding clarity issues but nonetheless implemented most of my detailed suggestions to improve the writing. This contradiction suggests either a reluctance to accept critique or a lack of clarity in their rebuttal. Please either accept critique or defend. This is more transparent.

Your research is valuable, and ensuring that the writing is as clear and precise as possible will increase its impact and reach. I suggest that the main authors consult the Nature guides on writing ([link 1] (<https://www.nature.com/articles/nphys724>), [link 2] (<https://www.nature.com/articles/nmeth.4532>), [link 3] (<https://www.nature.com/nature-portfolio/for-authors/write>)), which offer helpful advice for improving clarity and accessibility in scientific writing. These guides include relevant tips such as avoiding "elegant" variance and hyperbole. It was nice to see the authors removed most of the elegant variance from the initial submission (e.g. consistently using "CDR options").

Lastly, on reference styles: The issue is not that the manuscript was submitted in APA style, but that the referencing was done inconsistently. Defending this by noting that Nature will standardize the references later is insufficient. As I noted in my initial review, such carelessness is inconsiderate to reviewers, who rely on clarity and consistency to evaluate the manuscript effectively.

Remaining comments

1: In my first review I highlighted several mistakes that suggested carelessness and requested greater attention to internal revisions of the writing. That the main text of the revised manuscript starts with a typo and a misplaced comma "...reductions, need..." is not reassuring. Such oversights are inconsiderate to reviewers, who volunteer their time to improve the work. Please be kind to your peers.

The authors report finding very little research on BECCS, but this may result from not including broader research on bioenergy and CCS. While I have not verified this directly, I suggest that the authors provide a brief reflection on this potential limitation in the discussion.

I didn't receive a reply to my request for a critical reflection on the stark difference in results from the authors' initial search query (~70,000 articles) compared to two earlier reviews (2,900 and 6,200 articles in 2017 and 2018, respectively). The 1,000% increase in five years is striking and warrants exploration. A comparison of search strategies or reapplying earlier queries to recent data could offer alternative explanations for this difference.

While not a major issue, the authors provided a lacking reply to my comment starting with "In paragraph 8, it is argued". They reply that 10.5 days per author/research assistant (=366 workdays spread over 26 authors and 7 research assistants) for data collection would be an unwise use of their time. Many researchers, especially those conducting fieldwork or experiments, may find this perspective dismissive. This is particularly grating since you have funding for 7 research assistants. Less arrogance is always appreciated. As a side note, AI investments reflect monetary value to investors rather than social value, contrary to the implication in the manuscript.

To my comment starting with "In paragraph 3 it seems the authors aim to introduce a list of knowledge domains", the authors reply that it is unclear which paragraph is meant. This is a lazy and disingenuous reply, since the third paragraph after the "Introduction" (in the original submission) header says "CDR has been a key part of climate change mitigation discussions in the scientific literature, but has often been discussed in distinct knowledge domains".

The abstract reads "pointing towards potential inefficiencies at the science-policy interface". First and minor, this is unnecessarily vague. More importantly, after comments by the reviewer(s), the authors admit throughout the article that the number of studies should not necessarily be mirrored in policy. It seems the authors have forgotten their own improvement in this quoted sentence of the abstract.

Minor:

- "stabilising warming at or below" Why stabilize at 1.5°C if net-negative emissions are possible? This could be reconsidered, even if it was written as such in the most recent IPCC.

- "deploy CDR versus CDR's required role". This sentence sounds like massive use CDR is required by the real-world, but it primarily reflects the assumptions of model scenarios. Rephrasing could clarify this distinction.

- As noted earlier, a rate implies speed, and an increasing rate implies acceleration. "Ever increasing rates" would represent the third derivative ("jerk") in physics, which seems unlikely. "Increasing rates" would suffice for clarity.

- Most writing guides recommend avoiding exaggerated phrases like "great benefit to the research community" or "a strong focus," as they can weaken arguments rather than strengthen them.

Section 2.3 finds that 30 % of all CDR first-authors are from China, and 13 % is from the US. The Chinese and US populations constitute 17 % and 4 % of the world population. Without such context, readers may incorrectly conclude that China is disproportionately overrepresented, when the opposite is true.

"our intuition suggests that this proportion should be even greater". This statement seems unfitting, particularly after emphasizing that "ought ≠ is." Additionally, the term "this proportion" is unclear and could benefit from more specificity.

"A stronger focus on scenarios is supported". It is unclear whether this is a suggestion or a description. Clarifying the intended message would help readers.

"to be scaleable". This phrasing is confusing if the benefit has already been realized, as is the case, now that the study is

performed.

"diverse range of CDR options and interdisciplinary fields". I apologise that my initial comment was unclear. I meant to note that interdisciplinarity is different from multidisciplinary. In the revision, the authors make a valid argument, but it seems they may be referring to multidisciplinary (i.e., "researchers from different disciplines") rather than interdisciplinarity. If this is not the case, the phrasing is somewhat confusing, and additional clarification would help ensure the intended meaning is clear to readers.

Testament to the difficulty of clear writing, I was also unclear in my comment on the role of social science in evidence-based policy. When I said the humanities do not contribute to the evidence-base, it was not to say that humanities are irrelevant or without value, rather the opposite. The revision says "social sciences and humanities, for example to support evidence-based decision-making", but this misses my point. While the humanities offer valuable insights, they do not contribute to evidence-based policy because their contributions are interpretive rather than empirical. My intention was to highlight this distinction, not to diminish the importance of the humanities in broader decision-making processes.

(Remarks on code availability)

Version 2:

Reviewer comments:

Reviewer #5

(Remarks to the Author)

Dear Sarah Lück and co-authors

Thank you for your responses to my prior critiques, both about the tone of reply and the content.

The revised manuscript is clear, and let the findings speak on their own in modest language.

My remaining comments have been convincingly addressed, either by revision or fair rebuttals.

It was solid research, now it's also a solid paper. Congratulations!

(Remarks on code availability)

Reviewer 1

Comment	Response
This paper contributes a novel machine learning-assisted systematic mapping methodology to provide a comprehensive evidence map of the CDR literature. The manuscript is well-written and motivated. The written language is simple and easy to be read for people of a different background. The novelty of the study is also presented clearly and the systematic mapping approach is comprehensively presented in the supporting documents. Overall, a very useful and very much needed tool. With some additional filters this tool has the potential to provide a great service to the CDR community and climate change mitigation efforts in the wider context. However, I miss a reflection on the potential reason for certain results, which would be helpful for further guidance. Also, the discussion could benefit from clear recommendations for the CDR community on where to focus on (and why?). Also, it would be great to see a progression of the tool to more specificity (which might have been outside the scope of this work). For example:  • For BECCS, further coding based on biomass origin, for DACCS, further coding based on type of sorbent/solvent used, etc. This would allow an overview of the CDR literature space not only based on macro-CDR methods, but going more into the specific technology (e.g., high-temperature vs. low-temperature DACCS).  • For LCA studies categorize the function of the system (heat generation, biofuels production, application to soil, CO₂ sequestration), which would help in refining the literature landscape overview. 	Thank you for taking the time to review our manuscript and for helping us improve it. Regarding your main comments: First: We have provided additional reasoning to support our findings. The specifics can be found in the responses to your individual comments below. Regarding recommendations for the research community, any systematic map primarily highlights evidence hubs suitable for thorough systematic reviews and identifies evidence gaps where further research is essential to address societal needs. Both aspects have been added to the discussion section. Second: You offered many excellent suggestions for enhancing the automated sorting through machine learning, particularly by adding more layers to the sorting process or introducing new categories. While we appreciate the validity and potential value of each of these enhancements, they fall outside the scope of our current work and are not feasible to pursue. The reasons include:  1. Significant Additional Workload: Machine learning relies on hand-labeled data. Implementing your request would mean re-labeling around 6,000 documents, which previously took eight students about six months. This would considerably increase the workload, making it impractical within our current project scope. 2. Technical Challenges: Machine learning models require a substantial sample size to statistically evaluate patterns in text. However, some of the features you suggested are likely to have rather small sample sizes, which will limit our ability to build reliable models. Additionally, our model relies on data from the titles and abstracts of papers, and certain features—such as technical readiness level—are often not mentioned in these

	sections. As a result, these features cannot be effectively captured by the current machine learning approach. We are currently working on a living product, meaning that the data in this manuscript is constantly updated and made available. Over time we aim to constantly improve the data and might add some of the features you suggested. We are already working on additional classes of CDR options such as CCUS and Direct Ocean Capture. We will also invite other scholars to join this effort so we get an open-source product with community support. For now, if somebody wants to add the features you suggested we would argue that this can be effectively addressed through simpler methods than applying machine learning, such as keyword searches in titles and abstracts. With regards to the content of the paper: We added some reasoning behind some of the results. Recommendations for the research community coming from a systematic map mainly consists of points of evidence hubs which can be a source for a thorough systematic review, or pointing to evidence gap where we as society need more research to We added both parts to the discussion.
“In policy, CDR has gained increasing attention in recent years (Geden, Scott, and Palmer 2018; Schenuit et al. 2021), but many countries still lack concrete policies to scale CDR (Smith, S. M. et al. 2023) Chapter 5). This has led to a considerable gap between countries’ (so far limited) plans to develop and deploy CDR versus CDR’s required role in mitigation scenarios that stabilise global temperatures at an increase well below 2°C. (Bellamy 2018; Cox and Edwards 2019; Schenuit et al. 2021; Smith, S. M. et al. 2023). (page 2, main) o I would specify that the quantitative information of CDR required in mitigation scenarios is obtained from model-based scenario analyses. Hence it is a number resulting from modeling assumption and model “fingerprints.” I would find it important to specify that there is still a large spread of uncertainty associated with the scenario output: 1) Structural uncertainty (regarding technological innovation and uptake, market behaviour, etc. Assumptions in the models), 2) Parametric uncertainties (involving differences in parameter	We acknowledged your points here by adding the following sentence to the end of the cited paragraph: “One of the challenges here is that there is a large spread of possible CDR levels that countries might aim for, in part driven by model assumptions of technological innovation and potential market adoption (Dekker et al. 2023)”

calibrations, regional and temporal scale) and 3) Fundamental modelling choices (e.g., optimization versus simulation frameworks), model structure and foresight. (Dekker et al., 2023, Nature Energy)	
In Figure 3, it would be interesting to see, for the location-based research the correlations between CDR methods and the specific geographic locations studied. Is there a difference between this map and the results of Fig. 3a-3b?	In general, countries where lots of CDR research is produced, produce also lots of location-based CDR research. However, there is a difference when it comes to the methods studied in the different world regions. The reason for that is that we find a much higher percentage of nature-based solutions to be studied with a regional focus compared to other methods. This of course is then reflected in a different focus of the world regions. However, this result can be deduced from the results already presented in the paper. To accommodate your remark, we produced the same kind of figures which we used for 3a and 3b, present them in the SI and refer to them as following: “Further information on regional based research with details to the specific regions can be found in the SI.”
“In long-term mitigation scenarios that achieve the Paris long-term temperature goals (Byers et al. 2022) mainly BECCS (99%), afforestation (67%) and DACCS (29%) are the CDR options included. There is not a single scenario dealing with biochar or soil carbon sequestration, although these methods can deliver co-benefits such as food security or N₂O emission reduction (Lehmann et al. 2021).” Can we contextualize this statement more? Do the models not include biochar and soil carbon sequestration in their representation (i.e., the reason why these methods do not appear in the scenarios), or are they represented, but the models do not choose these methods in mitigation pathways (if so, why)?	It is the first case. The paragraph now reads “In long-term mitigation scenarios that achieve the Paris long-term temperature goals (Byers et al. 2022) mainly BECCS (99%), afforestation (67%) and DACCS (29%) are the CDR options included. There is not a single scenario dealing with biochar or soil carbon sequestration due to a lack of implementation of these methods despite their potential co-benefits such as food security or N₂O emission reduction (Lehmann et al. 2021).”
Coding Framework: Long-term storage is defined as at least 5 years. As 5-years is not a climate-relevant time horizon, why wasn't a longer time horizon chosen in the definition of long-term storage when screening for CDR literature?	We considered the 5 year horizon to be a good separator for dividing very short-term storage from insecure storage in nature-based carbon pools such as forests. We added this explanation in the coding guideline to make it clear for the outside. In reality, papers rarely describe the time horizon

	of storage in such a detailed way. It was rather used as a guideline to understand what we are interested in.
Figure 1 should be better explained. For example, the figure is not clearly explained in the text or caption (I know the methods are at the end in NC but this is not easy to follow and read to have this figure here). For example, it is unclear why and how documents are coded by hand. Why is each document coded by two people, does this mean all 30,000 papers?	We extended the figure caption. In Particular, we named the counts of each produced dataset to avoid the confusion you experienced. It now reads: Figure 1: Overview of the data retrieval for this study. Squares symbolise documents, a coloured square a document with labels, either assigned by hand (solid colour) or automatically (faded colour). Red documents are excluded, blue ones included. Step 1: 70,000 documents were retrieved from databases using search queries. Step 2: Of these about 6000 documents are sorted (=coded) by hand into being on CDR (relevant, blue squares) or being not on CDR (irrelevant, red squares). Documents on CDR were additionally described with CDR method and other categories. Step 3 and 4: The relevance labels and additional categories were used to train machine learning classifiers. Step 5: The trained classifiers were used to extend all labels to the unseen ~64,000 documents. Detailed information on methods can be found in the method section and the Supplemental Information.
The authors report the results consistently, however, I would be interested for the reason of reported results, e.g. why some CDR methods are more popular in certain regions. I would suggest to elaborate on this in either the results or the discussion. For example; ‘China focuses more on biochar research, Europe on BECCS, and North America, particularly the US, on DACCS relative to the research shares per technology in all countries. Research on ocean-based CDR methods is more predominant in Oceania and North America’ Can the authors give reasons why (in China)? For example, certain regimes, policies implemented, social pressure/acceptance? This would be useful to better understand how certain aspects can impact research efforts.	It is difficult to pin these trends on any of the given factors. More detailed explanatory analysis - itself a major undertaking - is also out of scope for this article. We therefore highlight a series of drivers in the text that are likely to influence publication rates, while acknowledging the importance of context: “There could be a number of drivers that explain the large uptake of biochar research in China, including institutional developments (e.g. increased core funding at agricultural universities, publishing incentives, or research grants), strengthening scientific networks (e.g. new societies, journals, project collaborations and exchanges), or a concerted push from the policy sphere (e.g. strategic research funding, support for public-private enterprises). Of course, applied research cannot be abstracted from its surrounding geographic and economic contexts. It is therefore not unexpected to find CDR research niches in different contexts (e.g. biochar

	in agriculturally productive regions, ocean based CDR in coastal regions, DACCS and BECCS in industrialised regions).”
I also do think that the results can be better structured in terms of paragraphs. Please think about the key messages you want to provide. For example, in section 2.2, the authors are going a bit back and forth between the number of studies of CDR options while I don't think that for each method one has to provide the results using its relative share of overall number of studies.	We enhanced the key message in section 2.2, i.e. that the attention to different CDR methods is very unevenly distributed. We further deleted several numbers from this paragraph and added a reference to the table where all the numbers are already noted.
If possible, please provide the search terms in the methods or SI.	The full search queries are already provided in an additional file. We added a reference to them in the method section.
Maybe I missed it, but why is storing CO2 in wooden buildings or concrete not considered as CDR option?	We developed a query related to CCUS, under which the storage of CO ₂ in wooden buildings is included. We also coded around 300 documents from this query. However, due to the complexity involved in storing CO ₂ in products (beyond just buildings), we decided not to include this CDR method in the paper. It will instead be addressed in a separate piece.
On the one hand, I liked the benchmarking of the research landscape with regard to the IPCC report. On the other hand, I am not sure how useful it is, i.e. I am not sure whether the IPCC should reflect the overall underlying scientific literature or instead should just choose the most relevant studies.	We agree that the distribution of IPCC citations does not need to represent the underlying literature. Still, we think that our comparison is interesting to highlight if a particular trend, such as the surge in biochar literature, goes unnoticed by the IPCC. We added the following sentences to the paper to acknowledge this: “Although it is clear that the IPCC cannot assess all of the large and growing body of available research (J.C. Minx, Callaghan, et al. 2017), it is essential to understand which main topics are emphasised or overlooked. We acknowledge that differences between the two literature bodies can arise from various factors. While we do not propose that the main topic distributions should necessarily align, our objective is to emphasise these differences and investigate their implications. This consideration reflects Hume's "is-ought" problem, which posits that observations about what is (the findings in the main literature body) do not inherently dictate what ought to be (the findings in the IPCC topic distributions).”
The discussion/conclusions seem more like a repetition of the results. They should be partly revised by provided a set of recommendations for	We re-wrote the discussion section substantially and specifically added a section on evidence hubs and evidence gaps to guide further research

the CDR community.	efforts.
It would be great if you mention somewhere in the paper, what makes this tool so useful and be specific about it. Highlight the added value of using machine learning but also the novelty aspect of your work.	The main advantages of using machine learning are that it allows us to analyze a much larger number of papers, making the map more comprehensive, and enables us to update the map with new literature as it becomes available. This point was previously mentioned in the first paragraph of the discussion, but we have now rewritten it for clarity. It now reads: “In this article we provide a comprehensive evidence map of the CDR literature. Our novel machine learning assisted approach follows a systematic mapping methodology (James, Randall, and Haddaway 2016; Saran and White 2018), and automates key labour-intensive parts of the process (Haddaway et al. 2020). This allows our systematic map to be scalable to the entire research domain around CDR rather than being limited to a single niche area of literature due to resource limitations (Nakagawa et al. 2018). As a result, we were able to quantify the CDR research landscape in an unprecedented way. Moreover, the automated classification can also be applied to newly published CDR research, representing a critical step forward in accelerating learning on CDR and providing high-quality evidence syntheses on the topic. This is particularly important, as we continue to be faced with a rapidly growing evidence base.”
About the tool: If possible, it would be great to have some additional filters in the tool itself; 1. Filtering the papers by research gap would make a major difference and position this tool above any existing online tool used for literature mapping as of today. It would probably mean a lot of work but think about the functionality of your tool. This would in turn increase its usability and expose your work widely in the research community and beyond. In addition to this, it would also be very interesting to have a discussion in the paper around what are the major research gaps being addressed as of today.	Thank you for this excellent suggestion. However, we believe the machine learning classification already offers substantial functionality by enabling filtering across all specified categories—such as CDR method, research methodology, and main context of the paper. Additionally, as proposed, papers can be further enriched with custom keyword searches. To enable filtering papers by research gap would require first identifying what ought to be researched – a task that cannot be undertaken by using our approach that focuses on what is already in the literature. Instead it would need the input of the entire research community on where knowledge gaps are. Our approach empowers researchers to tailor their searches according to specific research

	gaps, which are often highly dependent on the disciplinary perspective and individual researcher's interests. In our tool, researchers can easily locate the relevant papers corresponding to their particular area of focus, without aiming to provide a comprehensive ideal state of knowledge for the field.
2. In the LCA studies, one of the most important elements when putting together the study is the functional unit of the assessment which connects to the function of the system being analysed. For example, if I am looking at an LCA study of biochar, the system at hand could focus on the environmental impacts of biochar production. Other functions biochar is used for concerns: a) energy production, b) biofuel production c) biochar application to soil for carbon removal d) biochar used for soil purification etc. My point is, it is of crucial importance that you add another layer of filter where the LCA studies are filtered by the function of the system. For example, as of now the tool results in roughly 200 LCA studies on biochar but If I want to look only the studies on biochar application to soil, I will not get more than 6 to 10 LCAs. This would be a major innovative element of your tool.	As mentioned earlier, this is unfortunately out of scope. The same result could be more easily achieved through a simpler keyword search method. We added this thought to the discussion where we expanded the second paragraph "At the heart of our map of CDR research is a classification system trained with about 5,300 manually labelled documents that is able to predict not only the relevance of a scientific publication for the evidence map, but also the CDR method, the broad area of research and the research methodology applied." by these sentences "This literature base serves as a foundation for further analysis and can be easily expanded with additional features that provide more detailed descriptions of the scientific literature. In this context, a simple keyword search can enrich the literature landscape across a diverse range of interests."
3. In the same line as the LCA part, it is very important to provide a filter that specifies the IAMs scenarios used for the studies that work with them. And this is another point that can be discussed in your paper.	We hope we addressed this thought with the above comment.
4. Add another filter concerning the Technological readiness level (TRL) of the CDRs. In a way this could as well explain why there is a certain focus on specific CDRs compared to some others. In terms of functionality this would be very useful for different stakeholders beyond research (investors, governments) who want to learn mainly about the technologies that are ready for large scale implementation.	We liked the idea of reflecting the CDR technologies in the light of their TRL and added some sentences in the discussion.
Other minor comments:	

• Fig. 3 caption: “right” and “left” panel are inverted in the caption text	Done. Thank you!
• Data and Methods section, page 15, first paragraph: SI-Figure 5 instead of SI-Figure 4	We aligned the text and SI-Figure number. Thank you.
• Supplementary figures and numbered incorrectly: o E.g., SI Figure 1, related to Figure 3; instead of Figure 2	Done. Thank you!
• Can we not use machine learning in the title instead of using artificial intelligence, in the methods machine learning is used while in the title AI (I am aware that machine learning is a subset of AI though).	While we agree that "Machine Learning" is more accurate, the title length requires us to use an abbreviation. We believe "AI" is more widely recognized than "ML," so we have chosen to keep it in the title.
• I would refer to the online tool available on the web, somewhere in the paper. Is it open source and maintained?	We added a remark in the discussion and in the data availability section that all data can be downloaded online.
• I would suggest removing the last paragraph on page 4, the one that summarizes the findings in the introduction section. It is a great paragraph, but you mention these findings in the abstract, discussion and conclusion. I think it would help to reduce any sense of repetitiveness.	We replaced the result-like paragraph with an outlook of the article: “In this article, we first quantify the total volume of CDR literature and examine its temporal trends, as well as dissecting the literature by individual CDR methods to highlight shifts in research focus. Next, we investigate the origins of these studies, exploring regional profiles and analysing research that specifies geographic locations to identify patterns in CDR research distribution. We then assess the focus of the studies, including the scientific methods employed, to understand how research approaches have evolved. Additionally, we evaluate the representation of CDR literature in the recent IPCC report, comparing it to the overall CDR literature to highlight any discrepancies. Finally, we compare the attention given to different CDR methods across various contexts, including Integrated Assessment Model (IAM) scenarios, deployment strategies, and investment patterns. “

Reviewer 2

Comment	Response
I co-reviewed this manuscript with one of the reviewers who provided the listed reports. This is part of the Nature Communications initiative to facilitate training in peer review and to provide	Thank you for taking your time and helping us to improve our manuscript.

appropriate recognition for Early Career Researchers who co-review manuscripts.	
---	--

Reviewer 3

Comment	Response
I co-reviewed this manuscript with one of the reviewers who provided the listed reports. This is part of the Nature Communications initiative to facilitate training in peer review and to provide appropriate recognition for Early Career Researchers who co-review manuscripts.	Thank you for taking your time and helping us to improve our manuscript.

Reviewer 4

From the E-mail

Comment	Response
Carbon dioxide removal (CDR) and negative emissions technologies are needed in all scenarios that achieve global warming targets set by the Paris Agreement, and therefore are a crucial consideration for policy makers. This article utilizes emerging applications of artificial intelligence models to generate an automated systematic map of available academic evidence relevant for CDR. This allows the authors to characterize the distribution and extent of academic evidence across relevant metadata variables such as publication year, first author affiliation, intervention type, methodology and scientific discipline. In my view, an important and key contribution of this research is discrepancy that was revealed between the distribution of academic evidence, policy and deployments across different CDR technologies. This finding is particularly timely over the coming decade, as there a strong motivation to scale up CDR (particularly in novel technologies). This systematic map provides valuable insight into knowledge gaps that can inform future research priorities, as well as under-exploited knowledge clusters that can inform CDR investment and deployment.	Thank you so much for taking your time to review the manuscript and helping us to improve it. We provide responses to the specific recommendations in our comments below.

I want to congratulate the authors on this work, which represents a valuable application of how machine learning tools can be harnessed to conduct evidence synthesis with a large enough scope that is relevant for policy makers. I recommend that this article be accepted for publication, with revision. I have made most of my comments in the attached annotated pdf file, however include more general comments below. In particular, I think the discussion should elaborate more on key findings – specifically how patterns of CDR research differ compared to policy and practice, as well as providing more information and clarity in the methods with respect to the named entity recognition. For all the figures, please verify that the color palettes used are colorblind friendly where possible.	
Introduction	
- Many valuable points are presented in the introduction, however I think a few contours can be better defined that would help to position this work, its motivation and contribution – specifically with how increasing scientific literature interacts with trade offs between scope and available time resources when conducting systematic evidence syntheses, and how automated approaches can be applied in this context.	We edited the parts of the introduction, which introduce these ideas, for more clarity. For example, the point on the challenges of fast growing literatures now reads: “In the age of big literature (M. W. Callaghan, Minx, and Forster 2020; J.C. Minx, Lamb, et al. 2017; Nunez-Mir et al. 2016) - —where the scientific literature grows at ever increasing rates—balancing a research question’s scope and the resource demands for reviews, like reviewer time, is increasingly challenging (Collaboration for Environmental Evidence. 2018).. As a result, systematic mapping methodologies (systematic maps, evidence gap maps, etc.) have been developed by the evidence synthesis community” With regards to automation, we edited the corresponding sentences to now read: “Traditionally, systematic maps have been compiled manually and therefore are often limited in scope. Here, we use an approach that deploys machine learning methods to automate labour intensive tasks to provide an assessment at scale.” We hope that this improves readability.

- Many different types of interventions are discussed, and the inclusion of a typology figure or table that provides a short description of the main classes could be helpful to clarify the scope of interventions assessed in this map (essentially a summary of the table in the Coding Guideline – Categories – 0 Technology (CDR Technologies). This would also clarify the boundaries between the different types used in Figure 2 – e.g. blue carbon management can be a form/sub-type of soil carbon sequestration? I would suggest including a figure like this and moving table 2 to the supplement	Good idea! We followed your suggestion and hope you like the new figure 2 as much as we do.
Results	
- More precision/evidence could be added to several statements (e.g. briefly citing F1 statistics, confidence intervals, etc where relevant)	We cited classification performance and confidence intervals in the paper where we found it helpful and not too disturbing for the reading experience. Further, we extended the SI to include all confidence intervals.
- Section 2.1 & Figure 2: It seems for all the methods, that annual growth rate starts high and then declines, perhaps because it is easier to grow by a larger percentage when absolute numbers are low. So I wonder how useful a direct comparison is between growth rate for all publications in climate research vs CDR when growth rate seems so affected by the absolute size of the literature base? The overall size of the CDR literature base is given, but what is the size for all publications on climate change, and based on their differences in size, is a direct comparison in annual growth rates suitable?	Yes - thanks! We added “Growth in newly emerging research areas tends to be particularly high, as initial literature numbers are low, making each new publication a relatively larger addition to the existing body of work.” However, we would argue that this not true when the literature body reached a certain size, i.e. when comparing CDR literature and climate change literature.
Discussion	
See comments in main text file, particularly my recommendation to elaborate further on the disconnect between policy and practice and scientific knowledge	We rewrote the discussion section in large parts and included also your remarks.
Methods	
A note on data and code availability – will the raw data on the articles included in the study, and their predicted relevance for the different metadata variables, be provided along with source code (e.g. in an online repository)? This would be a valuable resource to the CDR community.	The complete data (documents and predicted categories) can be downloaded from our literature hub and we make the code available in a github-repository. We added links to these resources in the method section and also point now to the literature hub in the discussion as well.
I did not see detailed in the methods how the	True! It was accidently deleted. We added a

named entity recognition in titles and abstracts was performed – could you please specify?	citation to the main text and a sentence to the method section.
--	---

From the manuscript

Comment	Response
“from the atmosphere - potentially at very large scales”: replace with an 'em' dash	Done
“pull back global mean temperatures in an overshoot scenario;” - I think what is meant here is in the (likely) scenario where emissions would lead to an overshoot in the absence of CDR? But the way it could read is if an overshoot occurs, then CDR can be used. Also citation needed	It is actually meant the latter. We re-wrote the paragraph. It can be found in a very similar version in the recent IPCC report: “CDR has three distinct roles in climate change mitigation ((IPCC 2022) WG3 Chapter 12): first, reduce net CO ₂ and GHG emissions in the near term; second, to offset residual emissions from “hard to mitigate” sectors like industry, long-distance transport, and agriculture in the medium term (Buck et al. 2023; Luderer et al. 2018); and third, CDR to support sustained net-negative emissions in the long term, helping to lower global temperatures in overshoot scenarios and stabilising warming at or below 1.5°C ((IPCC 2022) WG3 Chapter 12). “
“CDR specifically in the land sector contributes to lowering net emissions today.”: Could cite the most recent State of Carbon Dioxide Removal report https://www.stateofcdr.org/	We deleted this part to make the introduction more concise.
“A stream of literature going back to the first IPCC assessment reports has considered the potential contributions of enhanced natural sinks through afforestation or soil carbon sequestration to achieve net emissions reductions”: I suggest providing a bit more clarity here, because the stream of literature about enhancing natural sinks is not limited to terrestrial nature based solutions -- although these might represent the initial interventions proposed within this category of interventions. Perhaps elaborating about how these were initial applications, but this area has since expanded to include analogous nature-based approaches within other biomes (e.g. coastal blue carbon) as well as more geo-engineering approaches that aim to	We added following sentence: “This area has since broadened to include analogous nature-based approaches in other ecosystems, such as coastal blue carbon (Nellemann et al. 2009), alongside options aimed at enhancing ecosystems' ability to absorb and store CO ₂ , like ocean fertilisation and macroalgae afforestation (Coale et al. 1998; N'Yeurt et al. 2012).”

enhance primary productivity (e.g. ocean fertilisation, macroalgae afforestation, etc).	
“discussed in the context of the integrated assessment modelling (IAM) literature”: This statement that BECCES is mostly discussed in a modelling context indicates that BECCS is not at the stage of development where it is being deployed -- if the subject of development stage/technological readiness is raised here then I think it should be briefly assessed for the other types of interventions mentioned.	The idea of this paragraph is to introduce the key CDR technologies and their chronological appearance in the literature. To make this clear we changed the sentence to: “Bioenergy with carbon capture and storage (BECCS) technologies have gained prominence in the early 2010s as an explicit option for achieving negative emissions in the integrated assessment modelling (IAM) literature [...]”
“such as ocean alkalinity enhancement (OAE)”: I would still classify this as enhancing natural sinks (see comment above). Can also reference: Oschlies, A., Stevenson, A., Bach, L. T., Fennel, K., Rickaby, R. E. M., Satterfield, T., Webb, R., and Gattuso, J.-P. (Eds.): Guide to Best Practices in Ocean Alkalinity Enhancement Research (OAE Guide 23), Copernicus Publications, State Planet, 2-oae2023, https://doi.org/10.5194/sp-2-oae2023 , 2023.	The emphasis in this paragraph is the development in time of the different CDR technologies. We therefore leave it as it is and added the reference you suggested.
“to synthesise these growing literatures”: To better define the contours of what has already been done, briefly summarise the scope of these syntheses -- and crucially what synthesis gaps remain that this synthesis addresses? Perhaps this just needs a statement to connect to the next paragraph about policy gaps	We added “but were either limited in scope or provide only broad overviews, making it challenging to draw specific conclusions” and added this text part later into the introduction.
“Smith, S. M. et al. 2023).”: Can also add the 2nd edition released in June 2024	Done.
“2016) - where”: em dash	Done.
“it is no longer possible for scholars to keep an overview of new scientific developments, even in specialised fields.”: This is a pretty strong statement -- fully manual systematic reviews and meta-analyses are still being published. However I think the point here is that as the amount of literature increases, the trade off between having a research question with a scope that is general enough to be scientifically interesting, and the resource requirements to conduct the review (e.g. reviewer time) becomes more difficult to balance as resource requirements increase (this is described in the Collaboration for Environmental Evidence Guidelines so you can cite this)	The sentence reads now: “In the age of big literature (M. W. Callaghan, Minx, and Forster 2020; J.C. Minx, Lamb, et al. 2017; Nunez-Mir et al. 2016)—where the scientific literature grows at ever increasing rates—balancing a research question’s scope and the resource demands for reviews, like reviewer time, is increasingly challenging (Collaboration for Environmental Evidence. 2018).”

“As a result, systematic mapping methodologies”: rather than claiming causality, perhaps it is more accurate to say that systematic maps can be an alternative to systematic reviews/meta-analyses to synthesize larger bodies of literature. Also they do not accomplish the same objective of a systematic review as they map the distribution of the evidence base and is therefore better suited to questions about identifying knowledge gaps and clusters as opposed to synthesizing effect sizes, etc.	To make the text more accurate we rewrote this section: “To address this issue, systematic mapping methodologies (systematic maps, evidence gap maps, etc.) have been developed by the evidence synthesis community (James, Randall, and Haddaway 2016; Saran and White 2018), to map existing literature, identify knowledge gaps and areas of abundance, and guide where reviews are most beneficial.”
“but they remain resource-intensive and have therefore been accompanied by discussions about the prospects of automation” : Connecting to my previous comment, I think they are also more amenable to automation because a systematic map does not require a validity assessment -- a stage in systematic reviews which requires a human to assess the bias of a research article	We also agree that systematic maps lend themselves to easier automation; however, we feel that this point is too detailed to include in this context.
“Systematic maps and evidence gap maps refer to a group of evidence synthesis methodologies that have been developed to comprehensively map out the available evidence on a particular research topic”: See my previous comments on systematic maps -- I think defining what a systematic map is should be positioned when they are first mentioned.	We deleted this text part as it is now mentioned already earlier in the discussion.
“Our study finds”: This paragraph may be better positioned in the Results section	We replaced the result-like paragraph with an outlook of the article: “In this article, we first quantify the total volume of CDR literature and examine its temporal trends, as well as dissecting the literature by individual CDR methods to highlight shifts in research focus. Next, we investigate the origins of these studies, exploring regional profiles and analysing research that specifies geographic locations to identify patterns in CDR research distribution. We then assess the focus of the studies, including the scientific methods employed, to understand how research approaches have evolved. Additionally, we evaluate the representation of CDR literature in the recent IPCC report, comparing it to the overall CDR literature to highlight any discrepancies. Finally, we compare the attention given to different CDR methods across various contexts, including Integrated Assessment Model

	(IAM) scenarios, deployment strategies, and investment patterns. “
“with high precision and recall,”: cite these metrics for the binary classifier	Done!
“total of 28,976 scientific”: Is a confidence interval available?	We added a confidence interval and added in the Method section and SI how we derived it.
“BECCS is volatile”: to support this, the standard deviation, or coefficient of variation could be provided	The standard deviation of the growth rate is 0.26 in the time between 2014 and 2022, for comparison: it is 0.05 for Biochar in the same time frame. However, we find this too distracting in the read and think that the figure 2, lower panel does illustrate the fact well enough.
“Right panel bottom: We sort the origin of the study into the world regions. For each world region we compare the percentage difference of the investigated CDR methods against all others from the complete dataset. Displayed are only the three highest and the three lowest differences. Left lower panel: Location-based research derived from locations mentioned in title and”: I think this is left panel bottom? Panle b?, Panel c?	Yes! We swapped left and right.
“we classified research methods and”: Clarify here that research method is from the multi-label classifier and not from the journal information, as using journal to infer research method is a fairly large assumption	We changed it to: “Our framework further enabled us to classify CDR research contents along key dimensions. In particular, we used our classifiers to distinguish research methods and the broad area of research. Additionally, we used journal information in combination with a research classification scheme to determine academic disciplines in line with the relevant OECD Category scheme”
“use multiple methods.”: Suggest to include a reference to SI Table 3	Done.
“Next, we analyse how the research landscape is reflected in the most recent 6th Assessment by the IPCC.”: Cite Fig 5	Done.
“only a small fraction”: provide the percentage?	It reads now: “We find that IPCC assessments are not a broad reflection of attention patterns in the underlying scientific literature on CDR methods. Overall, only a small fraction (2% of the CDR literature) of CDR studies are directly assessed.”
“cite a much lower share of reviews on CDR relative to the overall	In this form the sentence was wrong. Thank you for finding the error. We rewrote it:

literature,”: I am unclear what is meant here -- does this mean that the IPCC cites fewer reviews on CDR compared to reviews on other topics?	“While the IPCC includes a relatively higher proportion of reviews (19% vs. 15%) and systematic reviews (3% vs. 1%) compared to the overall CDR literature, our intuition suggests that this proportion should be even greater to help capture and provide access to the extensive uncited literature.”
“mainly BECCS (78%) and biochar (21%) - ”: Could this be added to the figure -- perhaps an additional bar for novel CDR deployments?	This figure already exists in the State of CDR, 1st edition. We added a citation to it.
“maps, which are assumed to be perfect”: Perhaps 'perfection' is a strong statement, however I think it would still be powerful (and more defensible) to say that they are commonly viewed as the 'gold standard' of evidence syntheses -- despite an arguable disconnect between transparency and reproducibility in the protocol, and how transparent/reproducible these reviews are in practise.	We changed the language as you suggested.
“The CDR literature is dominated by biochar research today – with a geographical centre in China.”: I suggest to briefly acknowledge some assumptions made by using first author affiliation to infer the origin of research -- authorship teams can represent diverse nationalities, and I wonder how well the distribution of first author affiliation data corresponds to the distribution of all authors? Also it is implied in this statement that China is leading the biochar research, when in many disciplines the last author can often be the 'leader' of the research, and furthermore 'funding' can still come from a different source. So there are several different factors which unfortunately we are unable to disentangle from the default metadata provided by citation-indexed databases like WOS and Scopus.	We added following part to address these limitations: “This approach simplifies the complexities of international collaborations, where authorship, lead roles, and funding often span multiple countries. However, we consider it a valuable proxy for drawing meaningful conclusions about the research origins.”
“due to its long-standing inclusion as a mitigation option under the UNFCCC and synergies with sustainable development and biodiversity. “: I think the finding of the disconnect between policy and practice and scientific knowledge is a key finding of this paper and warrants a longer discussion. For example, this difference could also arise because afforestation/reforestation interventions are relatively well-known (humans have been planting trees and managing forested landscapes for a long time) and therefore the best-practises and the co-benefits and	In addition to other edits in the paragraph, we now include the following sentence: “CDR deployment is also driven by issues like social acceptance, where methods with higher perceived “naturalness” and a longer history of practice (e.g. afforestation/reforestation) have a clear advantage (Osaka, Bellamy, and Castree 2021).”

dis-benefits might be better known compared to newer approaches. This could interact with factors such as social acceptance, which is likely higher for methods that are perceived as less invasive, to newer and more technology based interventions.	
“SI-Figure 4.”:SI Figure 5?	
“search strings with high levels of recall to make sure that as few scientific articles are missed as possible.”: Can you provide the date on which the search was conducted? Additionally, to verify that all types of CDR methods were represented by the validation dataset, can you provide the numbers of validation articles that are classified within each CDR type?	We added this information. We added a table to the supplementary material stating how many examples we had for each CDR technology in our validation dataset.
“Finally, we test the complete training strategy in a 3-fold cross validation”: Were the number of folds used to train the model different between the binary classifier for inclusion vs the metadata variables?	For all classifiers we used the same procedure. To make this clear we added: “Finally, we test the complete training strategy of all classifiers in a 3-fold cross validation” and also changed the wording in the SI to be more explicit on this point.

Reviewer 5

Comment	Response
Review report Submitted article: Scientific literature on carbon dioxide removal much larger than previously suggested: insights from an AI-enhanced systematic map Journal: Nature Communications Date of review: 3/7/2024 Synopsis The authors predict a much larger literature on CDR than previously found. They use a state-of-the-art supervised machine learning approach, but I am unable to judge if this is a methodical development, or just an advanced, but common, method applied to a new topic,	Thank you for taking the time to review our manuscript and helping us to improve it. Here, we present a comprehensive systematic map to clarify the current state of CDR literature. Systematic maps are a well-established, purely descriptive scientific method designed to identify both knowledge gaps and focal areas within a field (Saran and White 2018). The method has been recognized as valuable and published in high-impact journals (Berrang-Ford et al. 2021; Callaghan et al. 2021), underscoring their broad relevance. Currently, however, a complete systematic map of CDR literature is lacking, though it would provide researchers and

namely CDR. The two central findings of the paper are the large size of the CDR literature and the large focus on biochar. There are four findings of secondary importance: the affiliations of reviewed first authors are concentrated in China and OECD countries; most of the literature is categorized as technology-studies; the reviewed methods are mainly experimental or modelling-based; and the distribution of CDR types in the literature is different from that cited by IPCC and that in policy. Significance The application of machine learning is not a major scientific contribution on its own – and nor is it framed as such by the authors. If it was the main contribution, a method-oriented journal may be more fitting. When ML is used in a complementary fashion it can yield valuable insights, but the submitted version is mainly descriptive, and is only an incremental improvement to previous published reviews on CDR. While the authors prediction about the size and the distribution of the CDR literature are important and interesting findings, these are not put into play sufficiently to give further insights. The finding that biochar dominates the literature is interesting, but no empirical effort and almost no theorizing to understand what could explain this dominance. While the two main findings are important and of general interest, I believe they are incremental and that the paper in its current form is lacking an investigation into the results.	policymakers with a much-needed high-level overview of this fragmented research area. Machine learning in this study does not introduce a new method but rather supports the comprehensiveness of the systematic map, which we view as one of the paper's primary contributions. While still relatively novel in this context, with only a few existing examples, machine learning enables a level of scope and depth in mapping the literature that would be otherwise unachievable.
Recommendation Behind the paper is a substantial effort to screen and annotate a training dataset. The methods are rigorous. The explanation of the machine-learning approach is also easy to follow. The two central findings are of general interest. There are two main reasons I recommend a rejection of the paper in its current form. First, the major issue with clarity renders many arguments in the paper inaccessible. Many basic mistakes (referencing and formatting) coupled with unclear writing and multiple off-hand claims give the impression of insufficient internal editing. While I strive to separate form from content when giving critique, the lack of clarity detracts significantly from the overall quality of the paper.	Thank you for your feedback. While we disagree with your assessment regarding the style and clarity of the writing, we appreciate your acknowledgment of the methodological quality of our work. Your perspective on the writing is not aligned with the feedback from the other three reviewers, who specifically highlighted the style, quality, and clarity of our text. Nonetheless, we value your input and have adjusted the text where possible to enhance accessibility and clarity.
Second, while the two central findings are interesting, there is no investigation into potential	In response to this comment and others we have substantially re-written the discussion section to

underlying explanations. The attempts to contextualize the results are weak and occasionally based on a flawed premise. For instance, the two claims - that citation patterns in IPCC should mirror the distribution of CDR types in the predicted literature and that the distribution of literature point to evidence gaps on certain CDR types – are not convincingly justified. The secondary findings are of limited general interest, and without deeper analysis the central findings remain primarily descriptive.	contextualize our results. Please see more detailed responses below. We now also clarify that the comparison of the CDR literature to that which is cited in the IPCC does not imply that the IPCC ought to match this distribution. We want to emphasize again that systematic maps are to a large degree descriptive. However, such descriptives are relevant to understand what is going on in CDR research and require non-trivial scientific methodologies to derive. As a result, few efforts in this direction have been successfully published. New emissions accounting, another form of purely descriptive studies, are also regularly published in some of the high-level journals with a wide reach.
I encourage the authors to revise the paper thoroughly and resubmit to another journal, since the work contains important results that would be of general interest to the research community, and potentially, as the authors also mention, to the coming work of the IPCC cycle. Note: While I am confident to provide intermediate judgment, I am unable to provide an expert judgment of the machine learning approach. I have some, but not much, practical experience with applying supervised machine learning and transformer-based language models. From my judgement, the authors' methods are both rigorous and based in solid experience.	Thanks. The decision regarding whether we should submit elsewhere ultimately lies with the editor.
Main points	
The main argument of the paper is that the CDR literature is much larger than previously thought. This claim could be strengthened by testing whether the algorithm is overly inclusive i.e. predicts too many false positives. I understand the precision of the algorithm is measured on the training data, but further robustness could be checked by sampling some predicted documents. As of now the authors only test that the predicted documents are sufficiently inclusive (validation against a list of IPCC references). There might be advanced ways to do this, but it could also be as simple as extracting a random sample of documents and manually screening how many are relevant.	The purpose of a test set is to simulate the process of random sampling, which you mention when discussing the practice of "extracting a random sample." This approach isolates a portion of the labeled dataset from the training process entirely, enabling us to observe how the model performs on new, unseen data. Specifically, we employed a 3-fold cross-validation strategy, where the labeled dataset was split into three parts. In each iteration, two parts were used for training, while the remaining part served as the evaluation set. This process was repeated until each part had been used for testing, ensuring that we assessed our models on the full set of approximately 5,300 labeled samples. Additionally, we used an adjusted decision

	boundary which maximizes F1-scores. This means we set the threshold higher or lower to decide if a document should be regarded to belong to a class. Maximizing the F1-score means balancing precision and recall. Precision gets lower if we have a higher ratio of false-positive samples. This means that we used a strategy to prevent a high ratio of false positives. Importantly, this strategy was applied solely to the training data, with its effectiveness validated using the untouched test data, as described above. To further increase transparency on the uncertainties introduced by the machine-learning approach, our revised manuscript now also provides a confidence interval for the predicted size of the literature. With a deviation of up to 13% of the predicted value, this illustrates that our central claim is valid even when considering these uncertainties.
Figure 1 is well-designed and succeeds in conveying the machine learning approach.	Thank you!
A general issue is the vagueness of what constitutes a CDR study, and the subcategories. For instance, what does it mean that a document is classified as biochar research. The specific in-/exclusion criteria are found in the SI, but a general explanation in the main manuscript would be useful. The many ways in which classification is described leaves the reader unsure “Biochar research is covered in [x] publications”; “Roughly one third of CDR research ... refers to ...” “there is 3-4 times more research related to CDR” (this sound as if the literature is not mainly on CDR, but something else that is related); “CDR research addresses”; “the second largest category, SCS, with 23% of the total literature”; “BECCS represents only about 5% of all studies” (Side note: BECCS doesn’t represent any studies, it might be represented in some studies though); “the literature on CDR in general features mainly policy”. These examples raise questions of what it means that a study is on a type of CDR such as whether tangential attention sufficient for inclusion, is a mention in the title, abstract and keyword sufficient? What about an IAM-based study that includes, but is not focused on CDR? How many studies contain multiple CDR types? I don’t need answers to these specific questions here, but it would be a great improvement if such questions don’t arise when reading the	As the underlying issue here concerns the inclusion/exclusion criteria for studies, we now take steps to clearly elaborate this in the main manuscript. Specifically, we replaced Table 1 with an overview/excerpt of the coding guideline as was already suggested by another reviewer. This clarifies our labeling process and offers a clearer understanding of how studies are subsequently classified. Further, it is now clearly stated in the main manuscript that a core requirement for inclusion is that CDR methods need to be mentioned in the title, abstract or keywords of an article (a typical convention in review articles). This requirement may include or exclude IAM articles, depending on how much emphasis they place on CDR in their title/abstract/keywords. Additionally, we created an overview, presented in the SI, indicating how many studies were classified as addressing more than one CDR method or research approach. For a more detailed understanding we refer to the coding guideline where we clearly define how to include or exclude a study. We also corrected the sentence highlighted in the side note: “Bioenergy with Carbon Capture and Storage (BECCS) is represented in only about 5.6% of all

submission. I also suggest adding a clear definition of what is meant with a CDR study in the methods section, if not in the main. Of course, this can only be a general definition, that goes for all CDR types.	studies on CDR [...]"
The amount of predicted CDR literature is compared with previous scientometric studies, but it is unclear if the comparison accounts for the different time of the studies. Further, several of the authors have been part of previous reviews of CDR but found a lot less literature (Minx et al. 2018, Minx et al. 2017) in their search i.e. before screening. Since AI is only used to screen and not to search, it would be of general interest to explain why the initial search (unscreened) results in ~70,000 articles compared to ~6,200 in the 2018 study and 2900 in the 2017 study. Does this review cover more CDR types or were important keywords missing from the previous search queries? How much of the difference is explained by literature published after the previous studies?	We rewrote the sentence slightly to make it clear that we do actually compare the same time frame. Further, we provide the main reason for the discrepancy in the result section: "This is 3-4 times larger than what previous scientometric studies (J.C. Minx, Lamb, et al. 2017) or ongoing community efforts to manually track CDR research (Burns 2021) have suggested when comparing the same time range. For the former study, this discrepancy likely arises from their reliance on non-machine learning methods, which forced a high-precision, low-recall search approach. In the case of the manual tracking efforts, the rapid expansion of CDR literature has simply made comprehensive tracking unfeasible."
The authors emphasize multiple times that the literature cited by IPCC doesn't reflect the predicted literature: "IPCC assessments are not a broad reflection of attention patterns in the underlying scientific literature"; "Only 1% of the available scientific evidence on CDR was cited in the IPCC AR6", and "IPCC assessments cite a much lower share of reviews on CDR relative to the overall literature" (which is followed by the harsh and unfounded claim that this means that "IPCC therefore also fail to indirectly assess the available literature"). This emphasis is repeated for biochar, which dominates the predicted CDR literature, whereas IPCC is dominated by BECCS. However, it is not convincingly explained why IPCC citations should reflect the distribution of studies on each CDR types. In the same vein they write "understanding which subjects [CDR types?] receive more or less attention is vital" – This can be understood simply by looking at the IPCC references study i.e. for this, their mapping is unnecessary.	We think—and the other reviewers agree—that this section on the IPCC and the following section on the different indicators of industry and policy are of general interest and we therefore keep them in the main text. However, we agree that the topic distributions and attention patterns of the general literature and the literature cited by the IPCC do not need to be the same. We acknowledge this by rewriting the introduction to the IPCC section to: "Although it is clear that the IPCC cannot assess all of the large and growing body of available research (J.C. Minx, Callaghan, et al. 2017), it is essential to understand which main topics are emphasised or overlooked. We acknowledge that differences between the two literature bodies can arise from various factors. While we do not propose that the main topic distributions should necessarily align, our objective is to emphasise these differences and investigate their implications. This consideration reflects Hume's "is-ought" problem, which posits that observations about what is (the findings in the main literature body) do not inherently dictate what ought to be (the findings in the IPCC topic distributions)."

Continuing: “to ensure that IPCC reports continue to provide a balanced assessment of the evidence”. A balanced assessment of evidence connotes a pro-et-contra, but since this make little sense in this context, the reviewer is left to guess what the authors mean; should it be ensured that IPCC continues to cite CDR type proportionally to their distribution in the literature? (Side note: why say continue if IPCC doesn’t reflect it already, as was just claimed). I cannot see why such proportionality is important, but no arguments are given anyway. The authors seem to hold an underlying belief that x number of studies on CDR type A has a weight comparable to an equal number of studies on CDR type B – and therefore that the ratio (or relative number) of studies can indicate where more research is needed (e.g. “can help identify topical areas worthy of focus”; “we find substantial evidence gaps”). I am sceptical to the number of studies can be used in the way done by the authors, but even if so, it is a very noisy signal, and interpretation should therefore be cautious.	Our goal is not to prescribe focus areas for the IPCC, nor do we expect it to mirror the broader CDR literature exactly. However, if notable discrepancies arise—such as a substantially higher representation of biochar in the overall literature or a significantly stronger emphasis on certain scientific methods, like modeling studies in the IPCC citations—these differences warrant attention.
This critique above also applies to section 2.6, i.e. it is unclear why it is relevant or of value to compare how studies are distributed across CDR types with “other indicators of attention”. If this critique misses the mark in the way that the authors are not claiming that attention patterns should match, then it is still unmotivated why the authors want to compare patterns.	Although the data we present here derive from diverse sources—such as patent counts, capital investments, and the number of IAM scenarios—which may complicate noise assessment, the patterns are so markedly distinct that even variations influenced by noise would not alter the overall findings. Regarding your concerns about the general interest, other reviewers disagree, expressing that the work is indeed of broad relevance. In fact, they have requested even more detailed information. We further broadened the motivation by referring again to David Hume’s problem: “Finally, we find that the CDR options being researched most intensively are not the ones being most actively deployed, developed or invested in (Figure 7). Again, we do not imply that these distributions should necessarily be similar; rather, we aim to highlight and reflect on the differences between these categories”
The article compares the growth of CDR research to the growth of climate change literature, but it is not motivated why this is an	We introduce the comparison to the overall climate change literature as a benchmark.

interesting comparison, and I am unsure if it is a valid one. Research fields have highly different traditions of publication, e.g. because the cost and scale of experiments differ. I'm sure better ideas exist, but for a start a comparison with the mitigation literature or total scientific output across all disciplines would be improvements.	We use this benchmark because the data was readily available. While no benchmark is ever perfectly accurate, we think, it serves as a useful reference point. While we do not know of a easily accessible data to make a comparison with the mitigation literature, there is evidence that the entire scientific output is growing at around 5% per year (Bornmann, Haunschild, and Mutz 2021). This finding is consistent across several major literature databases and much smaller than the value we find for the CDR literature.
The three above mentioned critiques can also be related to David Humes famous "is-ought" problem. This issue was acknowledged in a previous publication by two of the co-authors (Callaghan, Minx & Forster 2020). This issue applies equally to this paper, although there is no acknowledgement of the issue, in this or in other terms.	This is indeed an important consideration that we now included in the manuscript. We acknowledge David Hume's "is-ought" problem in the paragraph on the IPCC literature and the one where we compare different attention markers.
The authors make an interesting note in mentioning that high biochar publication rates might be due to its potential co-benefits. Such remarks bring the authors knowledge about CDR to the front and is more relevant to a general journal such as Nature Communications than advanced scientometric methods.	Thank you.
In paragraph 8, it is argued that by using AI, the authors can assess the "balance of research" (unclear meaning) in terms of CDR types, research methods etc. But this can also be and is often done without AI. With 26 authors and 7 assistants in the acknowledgement, each would have to screen less than 4,500 articles (by titles first and abstracts only where necessary) and annotate less than 1600 documents (double-coding included). These numbers (70,000 and 29,000 divided by the number of co-authors and assistants) can be greatly reduced by first screening by journal subject areas, journal titles, and for irrelevant keywords in journal and document titles. This is not to say that there isn't a valid motivation for AI, but rather the one stated is insufficient.	Manually coding all 70,000 papers is neither scalable nor an effective use of human resources, as it demands an immense amount of time - approximately 366 full 8-hour workdays without breaks to classify each abstract in an average of 2 minutes. Instead, researchers' time would be far better spent on tasks requiring creative, analytical thinking. This trend is underscored by the substantial investments in AI-driven evidence synthesis by various research funders we see at the moment, highlighting the growing consensus on AI's value for handling such extensive datasets. Furthermore, our approach enables us to update the dataset on a regular basis without again screening and coding large sets of additional papers. We added this to the motivation in the introduction: "Furthermore, our machine-learning approach enables swift updating of the dataset in the future."
Generally the treatment of CDR is strangely uncritical although CDR is highly contested and politicized (see e.g. Anderson et al. 2023; Carton	Thank you for your thoughtful feedback on CDR, supported by numerous citations. In response to your remarks, we have removed the phrase

et al. 2023). Several of the co-authors have already contributed to this debate. To be more specific: The abstract and introduction emphasizes CDR as a critical strategy to limit global warming below 2°C. First, if any strategy is critical, it is emission reductions, whereas CDR is a supplementary strategy (Zickfeld et al 2023). CDR is meaningless if not combined or preceded by emission reductions (Fankhauser et al 2022). The authors mention three roles of CDR. The first role warrants further qualification, since even for residual emissions (hard-to-abate sectors), emission reductions are a preferable strategy to CDR in terms of several indicators (Fuhrman et al 2024). The second role hinges on a misguided fixation on the 2°C goal, rather than the objective of limiting global warming as much as possible. In other words, if emission reductions do not occur rapidly, reductions (and not CDR) will only become more urgent to minimize further global warming – which should be the target of climate change mitigation, rather than expecting CDR can be scaled to match current (i.e. unmitigated) emission levels. This fixation also ignores the expected (and those unknown) detrimental effects of temperature overshoot, even if temporary. The third role of CDR is indeed important, as some CDR options (land-based are mentioned) can lower net emissions and already does so (as mentioned by some of the co-authors in other publications). But any land-based CDR is limited by exactly that – land. In addition, climate change is not the only ecological crisis, which means there are other challenges to large-scale deployment of CDR beside the limited potential for removals. Many CDR options are found to have negative environmental effects (see e.g. Heck et al, 2018; Prütz et al, 2024; Wood-Hansen & van den Bergh 2024).

"critical role of CDR" and placed greater emphasis on the importance of emissions reductions. Additionally, we have reformulated the paragraph to closely align with the roles of CDR as outlined in the recent IPCC report.

It reads now in the abstract:
 "Carbon dioxide removal (CDR) will play an important role in any strategy to limit global warming to well below 2°C."

The text in the introduction reads:
 "To comply with the Paris agreement and to limit global warming well below 2°C rapid and deep GHG emissions reductions need to be complemented with Carbon Dioxide Removal (CDR) to actively remove CO₂ from the atmosphere—potentially at very large scales by mid-century and beyond (IPCC 2022; Smith, S. M. et al. 2023).

CDR has three distinct roles in climate change mitigation ((IPCC 2022) WG3 Chapter 12): first, reduce net CO₂ and GHG emissions in the near term; second, to offset residual emissions from "hard to mitigate" sectors like industry, long-distance transport, and agriculture in the medium term (Buck et al. 2023; Luderer et al. 2018); and third, CDR to support sustained net-negative emissions in the long term, helping to lower global temperatures in overshoot scenarios and stabilising warming at or below 1.5°C ((IPCC 2022) WG3 Chapter 12). Of course, CDR cannot compensate for stringent emission reductions, which need to be prioritised even in hard to mitigate sectors (Fuhrman et al. 2024). There are also deep uncertainties with respect to how fast CDR can be sustainably scaled-up, and whether the reversal of temperature overshoot can be safely achieved (Schleussner et al. 2024). This underlines the need to reduce emissions as fast as possible, while providing sufficient policy support that CDR can actually deliver gigatons of removals in the second half of the 21st century (Nemet et al. 2018; Smith, S. M. et al. 2023)."

Specific minor points

The authors mention they find little research on how to build CDR economies where the CDR (...) provides economic revenues beside mitigating global heating. This is a very specific example; an example of such literature would be interesting. It is however unmotivated why CDR should provide economic revenue beside mitigation – this seems a problematic requirement.	This part has been deleted on request by reviewers 1 and 4.
What is meant with a diverse range of interdisciplinary fields and why does it add to the complexity of reviewing literature? I would think interdisciplinarity would enable reviews, in contrast to unconnected fields.	We added some text to make the point clear: “The diverse range of CDR options and interdisciplinary fields involved in CDR research also adds to the complexity of this task, as researchers from different disciplines, each with their own specialised languages and methodologies, may be working on the same issues without fully knowing or engaging with each other due to misaligned terminology.”
It is mentioned that “there is little CDR research on the societal implications [of CDR] such as public acceptance, governance”. The choice of words here is imprecise and unfounded. The mention of societal implications is unclear – nowhere else are these mentioned. The reader is left to speculate if societal implications are understood as a subset of studies predicted to fall in the “context”-classification as “Policy / Government” or “Public perception”, but none of these equal “societal implications”. Note also that public acceptance specifically refers to public perception of policy after implementation (as opposed to acceptability which is ex ante policy adoption).	This paragraph is deleted on request by reviewers 1 and 4 to facilitate the reading.
The use of the Gini-coefficient to explain the distribution of the literature is unnecessary, as there is no reason to expect an equal distribution.	We deleted that part.
Section 2.4 refers to Keller et al. 2018 and Werner et al. 2022, but it is unclear what the role of these references is. Are they examples of studies under the label “policy and governance”, or are they meant to support a claim? If the former, please help the reader, for instance by writing, “see e.g.”. The same sentence claims that “policy and governance” and “earth system” are important categories but doesn’t explain why.	Indeed, these references serve as examples and we therefore added “see for example”. We deleted “important” in this sentence.
The authors find that the reviewed BECCS studies mainly fall into the “modelling” category	We deleted this overinterpretation of our results.

and infer that this “points toward limited efforts to research the design of the technology itself”. This inference is invalid, why would an increase of modelling studies using BECCS imply anything about other research efforts into BECCS?	
It is claimed that “There is not a single scenario dealing with biochar or ...”, but where does this claim from? Is it a result of the review?	This claim comes from a scenario database compiled for the 1st edition of the State of CDR report, which we cite in the corresponding figure and now also cite in the introduction of this section.
p. 13-14 states that the ML-assisted approach enables the authors to be systematic in their procedure, but as they also mention in the introduction there are also systematic reviews not relying on ML.	As stated above, we follow the systematic mapping procedure and complement it with our ML efforts. To make that clear we adjusted the sentence slightly: “The reason for this is that our machine learning assisted approach enables us to be systematic in our procedure and at the same time achieve both high levels of precision and recall.“ As explained above, previous approaches not using ML were simply less inclusive in their design of the search query, which is one of the main reasons why they find considerably less literature on CDR.
It is stated that DACCS is a place-specific CDR-type, but as I understand it, DACCS might be one of the least context-dependent types of CDR, except for the importance of local access to renewable energy? In the same paragraph is the trivial observation that there are fewer studies on novel technologies.	DACCS is indeed a place-specific CDR type, as it not only requires a substantial supply of renewable energy, as you noted, but also nearby access to storage facilities to avoid long-distance transportation of captured CO₂. For instance, the Roads2Removal project has conducted an analysis of local requirements for various CDR methods in the USA, highlighting the spatial and logistical considerations for DACCS implementation (see https://roads2removal.org/#direct-air-capture)
The authors call for more humanities and social science research to support evidence-based decision making. While I have sympathy for the point, the argument needs sharpening. First, the authors should be aware of the critique of the notion of evidence-based policymaking (see e.g. Head 2010). Second, the humanities do not contribute to evidence-based policymaking, nor is it the ambition.	We agree with Head 2010 that decision-making is complex, contested, and shaped by value preferences and practical judgements as much as scientific evidence. However, this does not mean that advocates for evidence-based policymaking should give up their voice in the competitive struggle for influencing decision-making. Indeed, to do so would leave the field wide open for interest groups that have a financial stake at play. Furthermore, social science and the humanities play a vital role by addressing essential research

	areas such as public perceptions, policy frameworks, governance, stakeholder engagement and equity concerns. By examining these aspects, they contribute a comprehensive reflection on policy making, encompassing both the policies themselves and the broader social context in which they operate. We edited the respective sentences to now read: “Additionally, our results point to a need for more research on CDR in the social sciences and humanities, for example to support evidence-based decision-making on questions of governance and equity. This type of research will be increasingly important as focus shifts towards the implementation of CDR at scale and policy design to support this, as is implied by the ambition of net-zero targets in the context of relatively slow action of mitigation policies.”
The authors suggest using their literature map to identify what has been “under-cited” by the IPCC. This is an interesting idea and sounds potentially useful. It could be an interesting addition to the paper. If not, please elaborate how this idea could be carried out.	We have adjusted the language and framing of the results that compare our literature map to the IPCC with regards to your previous comments. We refer to these for more detail (e.g. p. 24-25 of this document). Beyond this, additional analysis focusing primarily on citation patterns in the IPCC is out of scope for this article.
The authors make an unbacked and imprecise claim about what drives political attention (i.e. that relatively slow mitigation action will shift political attention toward CDR). What political attention - the media, the public, or that of political elites? It is entirely speculative that there should be a causal relation between a lack of emission reductions and an increasing attention toward CDR. If we are to speculate, the short-term prospects of economic growth and job creation through technology-exports associated with investments in CDR combined with the hope (among vested interests) that CDR can enable continued fossil fuel extraction are more likely drivers of elite political attention. How close we are to the 2°C goal is probably the least crucial factor in where political attention goes. The political economy of climate change is tightly associated with hegemonies and dominant processes of power i.e. capital accumulation.	We think this is a misunderstanding and have renamed section 2.6 to “2.6 Different shares of CDR options in other indicators of policy and practice”, explicitly avoiding the term “indicators of attention in policy and practice”.
Please provide colourblind-safe versions for the figures in which colour is the only way to distinguish elements. These can be put in the supplementary material. (See the Springer Nature commitment to inclusion)	Thank you for this overlooked topic. We provided alternatives where needed in the supplement and reference them in the figure caption.

On methods	
The main manuscript is lacking cut-off date for the search queries. The only place I can find it is in a figure in the supplementary material, but it is only given by the month.	We added the exact cut-off dates in the text.
In section 3 “our classification system [...] is not only able to predict the relevance”. Does it score of the degree of relevance of each study, or is it a binary sorting into relevant/irrelevant. Section 2.4 mentions the use of “journal information” without further explanation.	It can give a degree of relevance but we use it as a binary classifier. To make this clear we changed the text to: “that is able to predict not only if a scientific publication is relevant for the evidence map” The journal was used as a proxy for categorizing publications into research fields. We clarified this by editing the corresponding sentence: “Additionally, we used the journal, in which a publication appeared, to determine academic disciplines in line with the relevant OECD Category scheme [..]”
The difference between the annotation mentioned in paragraph 3 (screen and annotate) and 4 (we further annotate) is unclear.	We edited the paragraphs to make this difference clear and referenced figure 1. Paragraph 3 now reads: “We manually screen and annotate a total of 5,339 documents – 100-600 per CDR method – if they should be included in the map (distinction between blue and red squares in Figure 1)” Paragraph 4 reads: “In accordance with the definitions in our protocol (Lück et al. 2022), we further annotated all relevant scientific articles from our manually coded training and validation set with regard to the CDR options covered [... explanation of all annotated categories ...] These additional categories are represented in Figure 1 by the different blue shades for each annotated relevant document.”
The authors claim to achieve high levels of recall and precision. These terms are not my expertise, but I understand recall to be a ratio, but it seems here to be used in the sense of resulting in many search results. A footnote or explanation could be included in the supplementary material. The same goes for precision; it is unclear how it is measured. I understand these may be related to the classifiers and their scores such as F1 and ROC-AUC. Even though this is technical jargon, it can be explained in simple terms in the	Indeed, precision and recall are ratios that give the proportions of how many of the documents identified as relevant by the classifier were actually relevant and how many of those that are actually relevant were identified. We explain precision and recall in the main text and explain all terms in more detail in the SI.

supplementary material.	
The authors acknowledge that their classification system is not perfect, which is a trivial point. It is then argued that manually compiled evidence maps are assumed to be perfect, which sounds like an unnecessary strawman.	With this sentence we want to point out that manually compiled maps usually lack a validation strategy. Because another reviewer requested a change the sentence reads now: “However, our supervised machine learning procedures involve in-depth validation and as such we establish transparency about our uncertainty in quantifying the evidence base – something rarely provided in manually compiled evidence maps, which are commonly viewed as gold standard”
On form and style	
The lack of basic formatting in the supplementary materials should not need mentioning. Please provide page numbers in all documents. The lack of formatting in the document headed “Supplement” is not helping the reader (title, section headers, and subsection headers). All tables in the other supplementary material lack numbering and titles. For ease of reference, please add numbering the supplementary materials (e.g. SI 1, SI 2). The reference style is inconsistent in multiple ways, both in-text references and in the reference section. Examples can be found right next to each other, see for instance the top of p.3: (Jan C Minx et al. 2018; J.C. Minx, Lamb et al. 2017). The authors may have intended to clean such formalities up after receiving a journal decision, but this is inconsiderate to the reviewer.	In the SI we now  - provided page numbers - added a table of content with descriptive names - we reference each figure and table in the SI with SI figure 1, SI table 1, etc. in the figure/table caption - The citation style in Nature is numerical. We will change this once the paper is accepted.
The text is lacking clarity, which makes it hard to follow the intention of the authors in multiple places. The text needs a major edit of its writing. Some mistakes are basic, leaving the impression that the final text was not read from A-Z before submission. Some examples:	We generally do not agree with this harsh judgment which is also not supported by the other reviewers. However, we adjusted the text where we find this appropriate.
• Repeated sentences, one of which is used three times (including the references) in addition to the mention abstract. E.g. “CDR is a critical component of any strategy.”, “the CDR literature is 3-4 times larger than previously estimated (Burns 2021; J.C. Minx, Lamb, et al. 2017).”	The mentioned examples are no longer part of the text.
• Smith, S. M. et al. 2023 is referenced six times in the first four paragraphs. Another article (of the	Smith, S. M. et al. 2023 refers to a report, the State of CDR report with several chapters. We add

last author) is mentioned 8 times. Repeating a reference can be warranted, but not to the degree shown here.	chapter references to this reference to make it clear. The 8 times citation refer to J.C. Minx, Lamb, et al. 2017 which is actually a previous similar study. It is that often cited because we compare our - more comprehensive - study with the old one. We deleted one citation where this study is mentioned in a different context and appears alongside others.
• Figures are placed before they are introduced in the text.	We changed the text and figure order for Figure 1.
Unclear, unnecessary, mistaken and repetitive use of words. Some of these can be guessed by a specialist, but this should not be necessary. [Note that my formatting seem to have disappeared in the manuscripttrackingsystem from the below list of examples. The formatting was meant to indicate a line or word copy-pasted from the submitted article and my comment in parentheses] - e.g: important prerequisite (pleonasm); new scientific developments (tautology); actively remove CO2 (what is meant by active?) sustainably(?) scaling CDR; sustainably achieving gigatonne-scale; grows at ever(?) increasing rates; map out; methodologies vs. methods; “while research on other CDR methods” (are “other” those which are not land-based as mentioned earlier i.e. ocean-based?); Other biological CDR methods (the classification into biological CDR methods is not introduced previously); engineered CDR options (again, this is not an established category of CDR); growth rate of BECCS is...(BECCS doesn’t grow, the number of studies grow); Place-based research is important for ... in-situ; efficacy (effectiveness?), efficiency (what type?); provide an assessment at scale (“at scale” does not make sense); Most studies focus on investigating the CDR method from a technical perspective where the method itself is investigated; topical areas (are these simply topics?); distinctly different (unclear meaning); while patenting activity has been active in BECCS (most active or only active?); the literature is dominated by biochar research today (today -among articles published in 2024, or in the accumulated literature as indicated elsewhere?); with a geographical centre in China (institutional affiliations with China or studies in Chinese contexts?); most recent 6th Assessment	Thank you for your suggestions. We changed the wording in most of the suggested cases and only kept a few instances where we found them appropriate as they were.

Report (there is only one AR6); On p. 19, first section: research designs vary substantially across CDR methods (...) across individual CDR options research designs vary substantially (These sentences are in the same section, unclear if they mean something different?); please be clear on the use of compare with and compare to; we find further a strong focus on research on the methods (I assume the CDR literature focuses on methods, not research on methods); relative to the research shares per technology (this wording occurs several places and is very hard to understand); one third of CDR research in the literature (redundant); more place-specific research is mentioned in the same sentence as more specific sub-national locations (unclear if these terms mean the same); compared to vs. compared with.	
Section 2.4 mentions “Our framework”, but no framework is mentioned elsewhere.	We replaced “framework” with “ML approach”.
In methods section: ClimateBERT is chosen here due to its better performance (see Supplementary Information). Please be more specific when there are multiple supplementary materials with multiple sections.	We replaced “Supplementary Information” with “SI Table 2”.
In methods: summarized in Figure 1 and SI-Figure 4. The authors must be referring to SI-Figure 5.	Done.
• In the note to Figure 4: Please improve the descriptions of the panels, they are currently difficult to understand.	It is not clear what the reviewer finds hard to understand and none of the other reviewer had a remark on the figure caption. We slightly edited the caption to improve what we thought could be unclear.
There is no Table 1, and Table 2 is never introduced in the text.	We changed the numbering and added a reference to Table 1.
The panel descriptions of figure 3 doesn’t match the panel positions.	Adjusted.
When referencing a panel in a composite figure, it helps the reader if the reference specifies the panel (e.g. see Figure 3A).	We always refer to the complete figure for each paragraph.
What is meant by support in “a stronger focus on scenarios is supported by the fact that we find a shift from experimental research to modelling work”.	Is replaced by “underlined”.
Clarify when CDR in general refers to CDR in general and when it refers to a specific category.	Done.

The union of “Social science and economics” is the same as saying economics, since economics is a social science already. Do the authors mean social science or economics?	We replaced “and” with “including” since we still want to put some emphasis on economics.
In section 3: The sentence “our systematic map is scalable to the entire research domain around CDR” is confusing, as it sounds like the map isn’t scaled yet.	We edited the sentence as follows: “This allows our systematic map to be scalable to the entire research domain around CDR rather than being limited to a single niche area of literature due to resource limitations” to highlight the advantages of our approach compared to previous manual efforts.
The authors claim they perform the most comprehensive effort yet, but I assume something else than effort is meant.	We edited the sentence to now read: “While our CDR map represents the most comprehensive workeffort in this area to date, it does not offer a complete portrayal of CDR science.”
In paragraph 3 it seems the authors aim to introduce a list of knowledge domains. If so, please introduce the list properly and clarify when one list element ends, and another begins.	It is unclear which paragraph 3 is meant.
The abstract would benefit from more specificity. For instance, why not mention biochar instead of “very concentrated on specific CDR options”. The use of “policy and practice” as it stands in the abstract.	The parts in the abstract read now: “Growth in CDR research was faster than for the field of climate change research as a whole, but very concentrated in specific areas—such as biochar, certain research methods like lab and field experiments, and particular regions like China. Patterns of CDR research contrast significantly with trends in patenting and CDR deployment, pointing towards potential inefficiencies at the science-policy interface.”
In Figure 2, it is unclear if the middle panel shows accumulated shares, or the shares for each year of publication. It is hard to see the information presented in the bottom panel. Could it be enlarged by moving the top figure (less interesting) to the appendix, alternatively splitting the bottom figure into two; one representing the two lines “all publications on...” and another with the CDR types?	We added “per year” to make it clear. We find the above two panels more interesting and leave the figure as it is.
Figure 4, top panel. The axis label “publications mentioning technology” is confusing, especially given the text right before mentioned “technology research” as a category. If you mean “CDR method” (as used elsewhere), please be consistent.	We changed the axis label to “publications mentioning CDR option”.
Middle-panel: Many journals, such as Nature	We follow the OECD Category scheme for

Communications, are interdisciplinary journals. Such a “research field” seem to be missing from the diagram.	scientific journals (citation can be found in the text) which does not include “interdisciplinary journals”. We added the hint + citation also to the figure caption.
The types of CDR are sometimes referred to as CDR methods – this causes a tiny moment of confusion, as it could refer to the methods used in the CDR literature. I suggest avoiding the use of CDR methods and stick to other terms e.g. options, types, technologies, techniques.	We changed this terminology (previously “CDR methods”) to “CDR options” to avoid confusion with the methodology categories.
On the Supplementary material “Instructions for Coding” CDR-Syntra is described as a living database, but the use, purpose and accessibility of this database is not mentioned elsewhere. The “living”-feature is unclear to me, despite the elaboration (p1, list element 3); wouldn’t an automatically updated test dataset also require retraining the algorithm, as to ensure the training data is a valid sample of the complete dataset?	The “living” part, meaning that the CDR map will continuously be updated, is part of a larger project and not relevant for this paper. We deleted this part from the coding guideline. Further, the data is available on an online platform and this is now mentioned several times in the manuscript: In the discussion and the data availability section.
The readability of the many tables could be greatly improved with simple formatting, mainly to ensure that the tables are broken across pages as little as possible. Even if the authors planned to do such final formatting after the peer-review, it is considerate to the reviewers if such formatting is done before submission.	For the “Instructions for Coding” this is not feasible due to the length of the tables. Since all tables, except for the first one, follow a consistent format - always consisting of the same three columns ("Label," "Description/Definition," "Rule") - we believe this should still allow readers to follow the content without much difficulty. However, we ensured that all tables in the SI appear fully on individual sheets for better readability.
p. 19: The section “assessment” introduces a categorization of documents, which is not used in the manuscript. It would be an interesting addition, which could provide some of the lacking insight of what the large number of predicted studies focus on (and potentially into the constituents of the “technology” label).	We find that the data does not give more insight and excluded it from this study.
The “technology” label is very general, if not outright unclear. This is also evident from the large share of documents with a “main focus” predicted as “technology” (89 %). This warrants a decomposition of the label, otherwise it contains little informational value. The description of the label is described in the “main focus”-table (in “Instructions for coding”) as “the most	The “technology” label codes “Features, processes, and the application of the technology” - a category which is very common. Examples for that are studies which investigate optimal filters for DACCS, field studies of biochar to be used in agriculture or how mangrove afforestation affects fish habitats. The diversity already tells that there are many studies of this kind.

vague/catch-all label”, which is problematic since it suggest that coders assign difficult-to-classify documents to this category. This almost sounds as if the authors were already aware that this label could be improved from a re-naming or decomposition. Please do so.	Difficult-to-classify documents caused mostly a divergent labeling among the two coders which were first discussed by them and if they did not come to a conclusion we would discuss it in a larger group and - if necessary - added more rules to the coding guideline. A decomposition of the label technology is out of scope of this study because it would require a re-labelling of most of the classified studies, a process which would take an enormous amount of time as outlined earlier.
On the “Supplement”-file	
A table of contents would be helpful.	Done.
SI Figure 5: I suggest adding the number of manually screened and annotated documents to the figure, as to give the complete overview of the procedure.	The figure focuses on the data itself. The procedure itself is sketched in Figure 1. We do not see the need to repeat ourselves.
The number of “documents after deduplication” differ in the main manuscript and in SI figure 5 (75,518 vs. 69,942).	Adjusted.
Please give a brief introduction to the meaning of the indicators F1 and ROC and the the scoring scale (e.g. 0 to 1, where 1 means...).	We added an explanation in the SI.
References (only for those not already in the article) Anderson, K., Buck, H. J., Fuhr, L., Geden, O., Peters, G. P., & Tamme, E. (2023). Controversies of carbon dioxide removal. Nat. Rev. Earth Environ., 4, 808–814. doi: 10.1038/s43017-023-00493-y Brian W Head, Reconsidering evidence-based policy: Key issues and challenges, Policy and Society, Volume 29, Issue 2, May 2010, Pages 77–94, doi: 10.1016/j.polsoc.2010.03.001 Carton, W., Hougaard, I.-M., Markusson, N., & Lund, J. F. (2023). Is carbon removal delaying emission reductions? WIREs Clim. Change, 14(4), e826. doi: 10.1002/wcc.826 Fuhrman, J., Speizer, S., O’Rourke, P., Peters, G. P., McJeon, H., Monteith, S., ...Wang, F. M. (2024). Ambitious efforts on residual emissions can reduce CO2 removal and lower peak temperatures in a net-zero future. Environ. Res. Lett., 19(6), 064012. doi:	

10.1088/1748-9326/ad456d
 Prütz, R., Fuss, S., Lück, S., Stephan, L., & Rogelj, J. (2024). A taxonomy to map evidence on the co-benefits, challenges, and limits of carbon dioxide removal. *Commun. Earth Environ.*, 5(197), 1–11. doi: 10.1038/s43247-024-01365-z

Fankhauser, S., Smith, S. M., Allen, M., Axelsson, K., Hale, T., Hepburn, C., ...Wetzer, T. (2022). The meaning of net zero and how to get it right. *Nat. Clim. Change*, 12, 15–21. doi: 10.1038/s41558-021-01245-w

Heck, V., Gerten, D., Lucht, W., & Popp, A. (2018). Biomass-based negative emissions difficult to reconcile with planetary boundaries. *Nat. Clim. Change*, 8, 151–155. doi: 10.1038/s41558-017-0064-y

Wood Hansen, O., & van den Bergh, J. (2024). Environmental problem shifting from climate change mitigation: A mapping review. *PNAS Nexus*, 3(1), pgad448. doi: 10.1093/pnasnexus/pgad448

Zickfeld, K., MacIsaac, A. J., Canadell, J. G., Fuss, S., Jackson, R. B., Jones, C. D., ...Zaehle, S. (2023). Net-zero approaches must consider Earth system impacts to achieve climate goals. *Nat. Clim. Change*, 13, 1298–1305. doi: 10.1038/s41558-023-01862-7

References

- Berrang-Ford, Lea, A. R. Siders, Alexandra Lesnikowski, Alexandra Paige Fischer, Max W. Callaghan, Neal R. Haddaway, Katharine J. Mach, et al. 2021. “A Systematic Global Stocktake of Evidence on Human Adaptation to Climate Change.” *Nature Climate Change* 11(11): 989–1000. doi:10.1038/s41558-021-01170-y.
- Bornmann, Lutz, Robin Haunschild, and Rüdiger Mutz. 2021. “Growth Rates of Modern Science: A Latent Piecewise Growth Curve Approach to Model Publication Numbers from Established and New Literature Databases.” *Humanities and Social Sciences Communications* 8(1): 1–15. doi:10.1057/s41599-021-00903-w.
- Callaghan, Max, Carl-Friedrich Schleussner, Shruti Nath, Quentin Lejeune, Thomas R. Knutson, Markus Reichstein, Gerrit Hansen, et al. 2021. “Machine-Learning-Based Evidence and Attribution Mapping of 100,000 Climate Impact Studies.” *Nature Climate Change* 11(11): 966–72. doi:10.1038/s41558-021-01168-6.
- Saran, Ashrita, and Howard White. 2018. “Evidence and Gap Maps: A Comparison of Different Approaches.” *Campbell Systematic Reviews*. doi:10.4073/cmdp.2018.2.

Reviewer 1

Reviewer's comment	Response
Thank you for making the revisions based on the comprehensive set of comments. Although some of the comments cannot be entirely satisfied for various reasons, I believe the paper has been substantially improved. I would advise to accept the current version of the manuscript.	Thank you for your positive feedback and valuable input, which have significantly contributed to improve our paper.

Reviewer 2

Reviewer's comment	Response
I co-reviewed this manuscript with one of the reviewers who provided the listed reports. This is part of the Nature Communications initiative to facilitate training in peer review and to provide appropriate recognition for Early Career Researchers who co-review manuscripts.	Thank you for your time and effort in helping to improve our paper.

Reviewer 3

Reviewer's comment	Response
I co-reviewed this manuscript with one of the reviewers who provided the listed reports. This is part of the Nature Communications initiative to facilitate training in peer review and to provide appropriate recognition for Early Career Researchers who co-review manuscripts.	Thank you for your time and effort in helping to improve our paper.

Reviewer 4

Reviewer's comment	Response
I thank the authors for their diligent responses to my revisions and comments, and support the revised manuscript for publication. The additional elaboration on the role in advancing the frontier of AI-assisted systematic maps, as well as our understanding of the field of CDR better highlight the novelty and contribution of this work, and the addition of the new Figure 2 emphasizes the large perimeter of this synthesis.	Thank you so much for your valuable input. It really helped in improving the paper.

I feel that the revised document presents a clear explanation of the results, and well-supported interpretation and evaluation of their relevance. The authors made valuable additions for interpreting the accuracy and precision of their findings by including the performance and confidence intervals of the classifiers, as well as included numerous statements that better define the limitations of interpreting certain patterns and trends (e.g. high growth rates at low publication numbers in Fig 3). The revised section "2.5 IPCC reports, policy-making and practice have foci different from the scientific literature" enriches the comparison between the distribution of academic literature and publications informing policy. With regards to methodology, my previous request to add clarity with regards to the process for named entity recognition has been addressed.	
--	--

Reviewer 5

Reviewer's comment	Response
The authors have thoroughly revised the manuscript, and I now recommend acceptance given some minor revisions.	Thank you once again for taking the time to help improving our paper.
### Feedback on the reply to reviewer I found the tone of the reply document less than collegial and in places, replies appeared directed more toward the editor than the reviewer. This impression stems from several observations:  1. The authors repeatedly refer to other reviewers, which is irrelevant to my review process, as I had no access to their feedback. Replying to a comment is more convincing that referring to the judgment of another reviewer. 2. Phrases like _"this part has been deleted 	We sincerely apologize if we gave the impression that we do not value your specific input - this was absolutely not our intention. We deeply respect the scientific peer review process and, from our own experience, understand the significant effort required to provide constructive and valuable feedback. We are truly grateful for the time and effort you have dedicated to helping us improve. Thank you also for your detailed feedback on our comments. We appreciate your dedication and will take your suggestions into account in our future responses.

on request by reviewers 1 and 4" appear twice in reply to my comments. Acknowledging reviewer 5's feedback equally would have been appropriate. 3. Statements such as "the decision regarding whether we should submit elsewhere ultimately lies with the editor" are unnecessary and dismissive. This is self-evident, but reviewers are expected to make recommendations to the editor; I specifically used the term "encourage."

The authors seem more annoyed than appreciative of the peer-review process. This is disheartening given the voluntary nature of reviewing and the detailed feedback provided in my first review, which included examples and actionable corrections.

The authors note that they "generally do not agree with this harsh judgment" regarding clarity issues but nonetheless implemented most of my detailed suggestions to improve the writing. This contradiction suggests either a reluctance to accept critique or a lack of clarity in their rebuttal. Please either accept critique or defend. This is more transparent.

Your research is valuable, and ensuring that the writing is as clear and precise as possible will increase its impact and reach. I suggest that the main authors consult the Nature guides on writing ([link 1](<https://www.nature.com/articles/nphys724>), [link 2](<https://www.nature.com/articles/nmeth.4532>), [link 3](<https://www.nature.com/nature-portfolio/for-authors/write>)), which offer helpful advice for improving clarity and accessibility in scientific writing. These guides include relevant tips such as avoiding "elegant" variance and hyperbole. It was

nice to see the authors removed most of the elegant variance from the initial submission (e.g. consistently using "CDR options"). Lastly, on reference styles: The issue is not that the manuscript was submitted in APA style, but that the referencing was done inconsistently. Defending this by noting that Nature will standardize the references later is insufficient. As I noted in my initial review, such carelessness is inconsiderate to reviewers, who rely on clarity and consistency to evaluate the manuscript effectively.	
##### Remaining comments	
1: In my first review I highlighted several mistakes that suggested carelessness and requested greater attention to internal revisions of the writing. That the main text of the revised manuscript starts with a typo and a misplaced comma "...reductions , need..." is not reassuring. Such oversights are inconsiderate to reviewers, who volunteer their time to improve the work. Please be kind to your peers.	We apologize for this mistake and do not want to give the impression that we are careless. On the contrary, we try to address all your comments.
The authors report finding very little research on BECCS, but this may result from not including broader research on bioenergy and CCS. While I have not verified this directly, I suggest that the authors provide a brief reflection on this potential limitation in the discussion.	We believe there may be a misunderstanding. We classify any research as being on BECCS if it discusses CCS in conjunction with bioenergy as we have also fixed in the Coding Guideline. We are confident that the finding of limited research on BECCS accurately reflects the fact that comparatively little work has been conducted in this area.
I didn't receive a reply to my request for a critical reflection on the stark difference in results from the authors' initial search query (~70,000 articles) compared to two earlier reviews (2,900 and 6,200 articles in 2017 and 2018, respectively). The 1,000% increase in five years is striking and warrants exploration. A comparison of search strategies or reapplying earlier queries to recent data could offer alternative explanations for this difference.	We do not report on ~70,000 articles related to CDR but rather focus on ~29,000 articles. This difference arises from a distinct search query and search strategy. The mentioned previous approach did not apply filtering after running the search query. To ensure a high proportion of relevant documents (i.e., high precision) without additional filtering, they employed highly restrictive search queries, which may have excluded some relevant documents (resulting in low recall). In this study, we adopted a different approach using machine learning to deploy

	an additional filtering step. Initially, we used broader search queries to capture as many relevant documents as possible (high recall), even if this included many irrelevant ones (low precision), resulting in the ~70,000 documents you referenced. In the second step, machine learning was used to filter out irrelevant documents, allowing us to achieve both high recall and high precision. We believe this process is already adequately explained in the first paragraph of the results section of the paper: “There is a much larger body of CDR research than previously suggested. Based on our machine learning assisted approach that enables us to identify CDR studies with high precision (0.88 ± 0.0119, meaning the proportion of relevant studies among those identified is high) and recall (0.93 ± 0.005, indicating most relevant studies are captured) - see Methods, SI and Figure 1 - we predict a total of $28,976\pm 3800$ scientific studies in the Web of Science and Scopus (the two largest bibliographic core collections). This is 3-4 times larger than what previous scientometric studies (J.C. Minx, Lamb, et al. 2017) or ongoing community efforts to manually track CDR research (Burns 2021) have suggested when comparing the same time range. For the former study, this discrepancy likely arises from their reliance on non-machine learning methods, which forced a high-precision, low-recall search approach. In the case of the manual tracking efforts, the rapid expansion of CDR literature has simply made comprehensive tracking unfeasible.”
While not a major issue, the authors provided a lacking reply to my comment starting with "In paragraph 8, it is argued". They reply that 10.5 days per author/research assistant (=366 workdays spread over 26 authors and 7 research assistants) for data collection would be an unwise use of their time. Many researchers, especially those conducting fieldwork or experiments, may find this perspective	We truly believe that tasks requiring little to no intellectual or creative input should be automated whenever possible. The estimate of 10.5 working days per person is just that—an estimate. In reality, no one can sustain such work full-time. Personally, I find it impossible to engage in this type of task for more than 1–2 hours a day. Comparing this kind of automation to

dismissive. This is particular grating since you have funding for 7 research assistants. Less arrogance is always appreciated. As a side note, AI investments reflect monetary value to investors rather than social value, contrary to the implication in the manuscript.	fieldwork is not appropriate, as fieldwork inherently cannot be automated. We view our stance not as arrogant but as a wise allocation of intellectual resources. Moreover, automation offers the distinct advantage of being repeatable for newly published work, making it highly scalable. We have already addressed these points in our previous response and have adjusted the text in the document accordingly after our first response.
To my comment starting with "In paragraph 3 it seems the authors aim to introduce a list of knowledge domains", the authors reply that it is unclear which paragraph is meant. This is a lazy and disingenuous reply, since the third paragraph after the "Introduction" (in the original submission) header says "CDR has been a key part of climate change mitigation discussions in the scientific literature, but has often been discussed in distinct knowledge domains".	Since we addressed the reviewers' comments in order of their numbering, this section was deleted in a previous edit. We apologize for the inaccuracy in our earlier reply, which should have stated: "This part was deleted in a previous edit."
The abstract reads "pointing towards potential inefficiencies at the science-policy interface". First and minor, this is unnecessarily vague. More importantly, after comments by the reviewer(s), the authors admit throughout the article that the number of studies should not necessarily be mirrored in policy. It seems the authors have forgotten their own improvement in this quoted sentence of the abstract.	We replaced that part with: "highlighting the differing development stages of CDR technologies." to be consistent with the relevant paragraph on this in the discussion section.
Minor:	
- "stabilising warming at or below" Why stabilize at 1.5°C if net-negative emissions are possible? This could be reconsidered, even if it was written as such in the most recent IPCC.	We agree. "At or below" emphasizes that it is indeed possible to go below 1.5°C.
- "deploy CDR versus CDR's required role". This sentence sounds like massive use CDR is required by the real-world, but it primarily reflects the assumptions of model scenarios. Rephrasing could clarify this distinction.	We replaced "required" with "estimated".

- As noted earlier, a rate implies speed, and an increasing rate implies acceleration. "Ever increasing rates" would represent the third derivative ("jerk") in physics, which seems unlikely. "Increasing rates" would suffice for clarity.	Thanks, we implemented this suggestion.
- Most writing guides recommends avoiding exaggerated phrases like "great benefit to the research community" or "a strong focus," as they can weaken arguments rather than strengthen them.	We deleted "great" and "strong" in the given sentences.
Section 2.3 finds that 30 % of all CDR first-authors are from China, and 13 % is from the US. The Chinese and US populations constitute 17 % and 4 % of the world population, Without such context, readers may incorrectly conclude that China is disproportionately overrepresented, when the opposite is true.	We do not emphasize a conclusion that China is disproportionately overrepresented. Actually, we would prefer not to highlight this issue at all, as we don't know what a proportional representation of global research would look like, given that countries have different capacities and research foci.
"our intuition suggests that this proportion should be even greater". This statement seems unfitting, particularly after emphasizing that "ought ≠ is." Additionally, the term "this proportion" is unclear and could benefit from more specificity.	Thanks for this feedback. We have decided to refer back to one of the IPCCs stated goals, which is to comprehensively evaluate the available scientific literature. Thus, we changed the previous version: “While the IPCC includes a relatively higher proportion of reviews (19% vs. 15%) and systematic reviews (3% vs. 1%) compared to the overall CDR literature, our intuition suggests that this proportion should be even greater to help capture and provide access to the extensive uncited literature.” As follows: “While the IPCC includes a relatively higher proportion of reviews (19% vs. 15%) and systematic reviews (3% vs. 1%) compared to the overall CDR literature, we believe incorporating even more of these could further enhance its ability to fulfil a stated goal of the IPCC - which is to comprehensively evaluate the available evidence.”

"A stronger focus on scenarios is supported". It is unclear whether this is a suggestion or a description. Clarifying the intended message would help readers.	For clarity, we change it to: "The fact that IPCC assessments have tended to focus on scenarios is underlined by..."
"to be scaleable". This phrasing is confusing if the benefit has already been realized, as is the case, now that the study is performed.	"This allows our systematic map to be scalable to the entire research domain around CDR" is replaced by "This allows our systematic map to cover the entire research domain around CDR"
"diverse range of CDR options and interdisciplinary fields". I apologise that my initial comment was unclear. I meant to note that interdisciplinarity is different from multidisciplinary. In the revision, the authors make a valid argument, but it seems they may be referring to multidisciplinary (i.e., "researchers from different disciplines") rather than interdisciplinarity. If this is not the case, the phrasing is somewhat confusing, and additional clarification would help ensure the intended meaning is clear to readers.	We replaced "interdisciplinary" with "multidisciplinary".
Testament to the difficulty of clear writing, I was also unclear in my comment on the role of social science in evidence-based policy. When I said the humanities do not contribute to the evidence-base, it was not to say that humanities are irrelevant or without value, rather the opposite. The revision says "social sciences and humanities, for example to support evidence-based decision-making", but this misses my point. While the humanities offer valuable insights, they do not contribute to evidence-based policy because their contributions are interpretive rather than empirical. My intention was to highlight this distinction, not to diminish the importance of the humanities in broader decision-making processes.	We believe that the phrase "support evidence-based decision-making" implies that decision-making is not necessarily driven by the social sciences, but involves a mutual exchange between science and policy. We see no reason to distinguish the humanities in this process, as even interpretive evidence (for instance on the different values implicit in policy pathways) can be valuable to clarify different positions. Is it sufficient that we do not use the word "empirical" here?